# PepCompass: Navigating peptide embedding spaces using Riemannian Geometry

## Abstract

Antimicrobial peptide discovery is challenged by the astronomical size of peptide space and the relative scarcity of active peptides. Generative models provide continuous latent "maps" of peptide space, but conventionally ignore decoder-induced geometry and rely on flat Euclidean metrics, rendering exploration and optimization distorted and inefficient. Prior manifold-based remedies assume fixed intrinsic dimensionality, which critically fails in practice for peptide data. Here, we introduce **PepCompass**, a geometry-aware framework for peptide exploration and optimization. At its core, we define a **Union of $\kappa$-Stable Riemannian Manifolds** $\mathbb{M}^\kappa$, a family of decoder-induced manifolds that captures local geometry while ensuring computational stability. We propose two local exploration methods: **Second-Order Riemannian Brownian Efficient Sampling**, which provides a convergent second-order approximation to Riemannian Brownian motion, and **Mutation Enumeration in Tangent Space**, which reinterprets tangent directions as discrete amino-acid substitutions. Combining these yields Local Enumeration Bayesian Optimization (**LE-BO**), an efficient algorithm for local activity optimization. Finally, we introduce Potential-minimizing Geodesic Search (**PoGS**), which interpolates between prototype embeddings along property-enriched geodesics, biasing discovery toward seeds, i.e. peptides with favorable activity. *In-vitro* validation confirms the effectiveness of PepCompass: PoGS yields four novel seeds, and subsequent optimization with LE-BO discovers 25 highly active peptides with broad-spectrum activity, including against resistant bacterial strains. These results demonstrate that geometry-informed exploration provides a powerful new paradigm for antimicrobial peptide design.

## 1 Introduction

Efficient exploration of peptide space is notoriously difficult. At the global level, there are more than $3.3 \times 10^{32}$ combinatorially possible amino acid sequences of length at most 25. At the local level, each peptide of length 25 has nearly 1000 neighbors within an edit radius of one. Moreover, only a small fraction of amino acid sequences correspond to antimicrobial peptides (AMPs), which have high activity against bacteria Szymczak & Szczurek (2023); Szymczak et al. (2025). This extreme combinatorial complexity renders AMP discovery by brute-force exploration intractable. To address this challenge, we turn to one of the most prolific inventions of humankind: maps.

Since the dawn of civilization, maps have provided a structured way to support both local and global navigation, driving scientific discovery. In modern machine learning, latent-space generative models such as VAEs, GANs, WAEs, and normalizing flows (Bond-Taylor et al., 2021) enable building continuous latent representations—maps—of peptides. Such maps have already facilitated the discovery of promising new AMPs (Szymczak et al., 2023; Oort et al., 2021; Wang et al., 2022; Das et al., 2020). The standard workflow assumes that once the model is trained, its latent space together with the decoder properly models a set of valid, synthetizable peptides. The latent space is typically chosen to be $\mathbb{R}^d$ with a flat Euclidean metric, enabling direct application of existing exploration and optimization algorithms. However, such flat representations suffer from a significant flaw: they ignore the differential geometry induced by the decoder, leading to distortions in distances.

Typical approaches attempting to circumvent this problem assume the *manifold hypothesis* (Bengio et al., 2012) and use the pullback metric (Arvanitidis et al., 2018). However, recent work (Loaiza-

Ganem et al., 2024; Brown et al., 2022; Wang & Wang, 2024) has shown that the manifold hypothesis does not withstand empirical scrutiny for image data, where sets of images are better modeled as unions or CW-complexes of manifolds with varying low dimensionality (we refer to Appendix A for Related Work). We show that peptide spaces suffer from a similar issue and introduce decoder-derived **Union of $\kappa$-Stable Riemannian Manifolds** $\mathbb{M}^\kappa$, which captures both the complex structure of peptide space and its local geometry, with computational stability controlled by a parameter $\kappa$. Intuitively, we cut the globally distorted map into a set of charts that enable efficient and distortion-free exploration and optimization.

Building upon the union-of-manifolds structure, we introduce **PepCompass**, a geometry-informed framework for peptide exploration and optimization at both global and local levels. At the *global level*, we propose Potential-minimizing Geodesic Search (**PoGS**), which models geodesic curves between known prototype peptides to identify promising seeds for further optimization (Figure 1A). We represent geodesics as energy-minimizing curves in peptide space and augment them with a potential function encoding antimicrobial activity. This biases exploration toward regions not only similar to the starting prototypes but also exhibiting higher activity. In doing so, our method extends standard local analogue search around a single prototype into a bi-prototype, controllable regime.

For *local* search on a single manifold from the family $\mathbb{M}^\kappa$, we designed two geometry-informed approaches: Second-Order Riemannian Brownian Efficient Sampling (**SORBES**) and Mutation Enumeration in Tangent Space (**MUTANG**). SORBES is a provably convergent, second-order approximation of the Riemannian Brownian motion (Schwarz et al., 2022; Herrmann et al., 2023), serving as a Riemannian analogue of local Gaussian search. MUTANG addresses the discrete nature of peptide space by reinterpreting the local tangent space not as continuous vectors but as discrete mutations, directly corresponding to amino-acid substitutions. This reinterpretation provides both interpretability and efficiency, enabling enumeration of a given peptide's neighbours. We further combine SORBES and MUTANG into an iterative **Local Enumeration** procedure that densely populates the neighbourhood of a given peptide with valid, diverse neighbours. Finally, integrating this enumeration with a Bayesian optimization scheme yields an efficient Local Enumeration Bayesian Optimization procedure (**LE-BO**; see Figure 1B,C).

*In vitro* microbiological assays demonstrated unprecedented, 100% success rate of PepCompass in AMP optimization. Using PoGS we derived four peptide seeds, all of which showed significant antimicrobial activity. Further optimization of these seeds with LE-BO yielded 25/25 highly active peptides with broad-spectrum activity, including activity against multi-resistant bacterial strains. Code is available at `https://anonymous.4open.science/r/pep-compass-2ABF`.

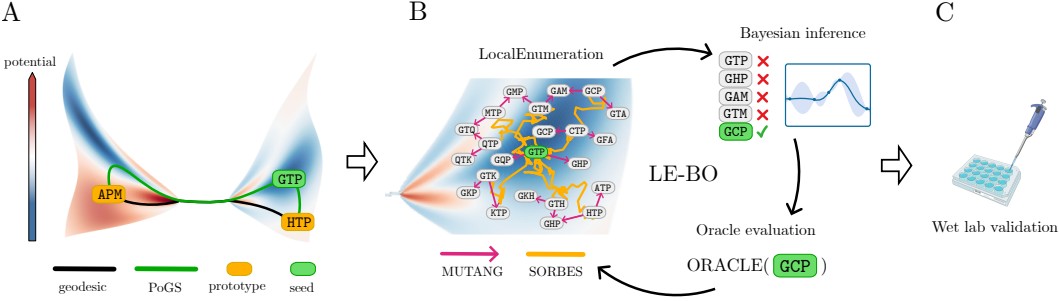

Figure 1: **PepCompass overview.**

## 2 METHODS

### 2.1 BACKGROUND

Let $\mathrm{Dec}_\theta \in C^\infty : \mathcal{Z} \to \mathcal{X}$ be the deterministic decoder mapping latent vectors $z \in \mathbb{R}^d$ to position-factorized peptide probabilities $\mathrm{Dec}_\theta(z) \in \mathbb{R}^{L \times A}$ (with $L$ the maximum peptide length and $A$ the size of the amino acid alphabet $\mathcal{A}$ extended with a padding token $pad$, and $\mathbb{R}^{L \times A}$ - set of matrices of shape $(L, A)$). Define

$$X = \mathrm{Dec}_\theta(\mathcal{Z}) \subset \mathbb{R}^{L \times A}, \qquad \mathrm{p}(z) = \mathrm{argmax}(\mathrm{Dec}_\theta(z), \dim = 1) \in \mathcal{A}^L,$$

where $\mathsf{p}(z)$ denotes the decoded peptide sequence. Intuitively, $X$ is a continuous probabilistic approximation of the peptide space (with $\mathcal{Z}$ as its map), from which the concrete peptides are decoded back using the $\mathsf{p}$ operator. For clarity, we drop explicit parameter dependence and simply write Dec instead of $\text{Dec}_\theta$. We additionally introduce $\hat{\text{Dec}} : \mathcal{Z} \to \mathbb{R}^{LA}$ and $\hat{X} = \hat{\text{Dec}}(\mathcal{Z})$, i.e. the flattened versions of Dec and $X$.

A common approach is to equip $\mathcal{Z}$ with the standard Euclidean inner product $\langle \cdot, \cdot \rangle_{\mathbb{R}^d}$, and use the associated Euclidean distance as a base for exploration. However, this ignores the geometry induced by the decoder, which, under the manifold hypothesis, can be accounted for using the *pullback metric* (Bengio et al., 2012; Arvanitidis et al., 2018).

**Pullback metric**   Under the manifold hypothesis, $\hat{\text{Dec}}$ is full rank (i.e. $\text{rank } J_{\hat{\text{Dec}}}(z) = d$ for all $z \in \mathcal{Z}$, where $J_{\hat{\text{Dec}}}(z) \in \mathbb{R}^{(LA) \times d}$ is the decoder Jacobian), and $(\mathcal{Z}, G_{\hat{\text{Dec}}})$ is a $d$-dimensional Riemannian manifold (do Carmo, 1992) where $G_{\hat{\text{Dec}}}$ is a natural, decoder-induced pullback metric on $\mathcal{Z}$ given by:

$$G_{\hat{\text{Dec}}}(z) = J_{\hat{\text{Dec}}}(z)^\top J_{\hat{\text{Dec}}}(z) \in \mathbb{R}^{d \times d}, \tag{1}$$

for $z \in \mathcal{Z}$. To simplify notation, let us set $G_{\text{Dec}} = G_{\hat{\text{Dec}}}$. Now, for a tangent space $T_z \mathcal{Z}$ at a point $z$ and tangent vectors $u, v \in T_z \mathcal{Z}$,

$$\langle u, v \rangle_z^{\text{Dec}} = u^\top G_{\text{Dec}}(z)\, v. \tag{2}$$

Note that $\hat{\text{Dec}} : (\mathcal{Z}, G_{\text{Dec}}) \to (\hat{X}, \langle \cdot, \cdot \rangle_{\mathbb{R}^{LA}})$ is an isometric diffeomorphism.

**When the manifold hypothesis fails: union of manifolds**   Previous work demonstrates that the manifold hypothesis often fails for complex data such as images (Loaiza-Ganem et al., 2024; Brown et al., 2022; Wang & Wang, 2024), and the data is better represented as unions of local manifolds of varying dimension, typically lower than that of the latent space. However, the previous methods defined the submanifolds based on pre-specified datasets and could not generalize to new data.

## 2.2   Union of $\kappa$-Stable Riemannian Manifolds

Assuming that the manifold hypothesis is indeed violated and that a given generative model has learned to faithfully capture the lower-dimensional structure in the data, it should be reflected in the decoder having rank strictly smaller than the latent dimensionality. We verified this phenomenon for antimicrobial peptides data in two state-of-the-art latent generative models (Das et al., 2018; Szymczak et al., 2023) (see Appendix B). This implies that the $G_{Dec}$ is not of full rank, and consequently the pair $(\mathcal{Z}, G_{\text{Dec}})$ does not constitute a Riemannian manifold.

To address this, we equip each point $z \in \mathcal{Z}$ with a local, potentially lower-dimensional manifold, which we further enrich with a Riemannian structure from the pullback metric. In contrast to previous methods, we adapt a decoder-dependent approach in the submanifold definition, enabling generalization to any point encoded in the latent space. Namely, we decompose $\mathcal{Z}$ as a union of locally $\kappa$-stable Riemannian submanifolds ($\kappa \geq 0$)

$$\mathbb{M}^\kappa = \{M_z^\kappa : z \in \mathcal{Z}\}, \qquad M_z^\kappa = \left(W_z^\kappa, G_{\text{Dec}}\right),$$

where each $W_z^\kappa \ni z$ is an open affine submanifold of $\mathcal{Z}$ of *maximal dimension* (denoted as $k_z^\kappa$ and refered to as $\kappa-$stable dimension), such that the pullback metric $G_{\text{Dec}}$ restricted to $W_z^\kappa$, denoted $G_{\text{Dec}}|_{W_z^\kappa}$, has full rank and satisfies the $\kappa$-*stability condition*

$$\inf_{\substack{v \in T_z W_z^\kappa \\ \langle v, v \rangle_{\mathbb{R}^d} = 1}} \langle v, v \rangle_z^{\text{Dec}} > \kappa^2.$$

Intuitively, $W_z^\kappa$ removes degenerated (non-active) directions of the decoder, ensuring that all eigenvalues of $G_{\text{Dec}}|_{W_z^\kappa}$ are bounded below by $\kappa$. This guarantees numerical stability for geometric computations requiring inversion of $G_{\text{Dec}}$. Note, that similarly to the full-rank case, $\hat{\text{Dec}}|_{W_z^\kappa} : (W_z^\kappa, G_{\text{Dec}}) \to (\hat{\text{Dec}}(W_z^\kappa), \langle \cdot, \cdot \rangle_{\mathbb{R}^{LA}})$ is an isometric diffeomorphism.

The explicit SVD-based construction of $W_z^\kappa$ is deferred to Appendix C. For stability near boundaries, we also use contracted domains $W_z^\kappa(\alpha) = \{z + \alpha(v - z) : v \in W_z^\kappa\}$ and $M_z^\kappa(\alpha) = (W_z^\kappa(\alpha), G_{\text{Dec}})$, with $\alpha \in (0, 1)$.

## 2.3 SORBES - Second-Order Riemannian Brownian efficient Sampling

Having a stable Riemannian approximation $M_z^\kappa$ of $X$ in the vicinity of a point $z \in \mathcal{Z}$, we now describe how to explore it efficiently within the local neighbourhood of $z$. Our goal is to simulate a Riemannian Brownian motion for a time $T$ starting from $z$ that is a Riemannian equivalent of a local Gaussian perturbation $z + \epsilon$, $\epsilon \sim \mathcal{N}(0, T)$. To this end, we introduce the SORBES (Second-Order Riemannian Brownian efficient Sampling) procedure, described in Algorithm 1. SORBES improves the flat Gaussian noise by exploring only active local subspace $W_z^\kappa$, it is isotropic w.r.t. to the decoder geometry, and respects the local curvature of $M_z^\kappa$.

---

**Algorithm 1** SORBES

---

**Require:** $z \in \mathcal{Z}$, $\kappa \geq 0$, step size $\epsilon$, diffusion time $T$, $\alpha = 0.99$
1: Initialize $z_0^\epsilon \leftarrow z$, stopped $\leftarrow$ False, $\sigma \leftarrow 0$          ($\sigma$ tracks diffusion time)
2: $W_z^\kappa, G_{\text{Dec}} \leftarrow M_z^\kappa$
3: **for** $i = 1$ to $\lfloor \frac{T}{\epsilon^2} \rfloor$ **do**
4:     Sample a unit tangent direction $\overline{v} \in S_z^\kappa = \{u \in T_z M_z^\kappa : \langle u, u \rangle_z^{\text{Dec}} = 1\}$
5:     Set $v \leftarrow \sqrt{k_z^\kappa}\, \overline{v}$
6:     **if** not stopped **then**
7:         Update ($\Gamma$ denotes the local Christoffel symbol for $M_z^\kappa$)

$$z_i^\epsilon = z_{i-1}^\epsilon + \underbrace{\epsilon v}_{\substack{\text{first-order} \\ \text{geodesic approximation}}} - \underbrace{\epsilon^2 \Gamma(z_{i-1}^\epsilon)[v, v]}_{\substack{\text{second-order} \\ \text{geodesic approximation}}},$$

8:         $\sigma \leftarrow \sigma + \epsilon^2$          (diffusion time update)
9:         **if** $z_i^\epsilon \notin W_z^\kappa(\alpha)$ **then**
10:           stopped $\leftarrow$ True
11:         **end if**
12:     **else**
13:         $z_i^\epsilon \leftarrow z_{i-1}^\epsilon$          (absorbed state)
14:     **end if**
15: **end for**
16: **return** $(z_i^\epsilon)_{0 \leq i \leq \lfloor \frac{T}{\epsilon^2} \rfloor}$, $\sigma$

---

Before introducing the key theoretical property of this algorithm, let's recall the crucial notation. For $A \subset M_z^\kappa$, $A^c$ is the complement of $A$ in $M_z^\kappa$. Let $d_{M_\kappa}$ be the geodesic distance on $M_z^\kappa$ w.r.t. to the pullback metric, and $d_{M_z^\kappa}(x, A) = \inf_{y \in A} d_{M_z^\kappa}(x, y)$ for $x \in M_z^\kappa$ and $A \subset M_z^\kappa$. Let Ric be the Ricci curvature. Then the key theoretical property of the Algorithm 1 is summarized by:

**Theorem 1.** *Let $(Z_i^\epsilon)_{i \geq 0}$ be the sequence produced by Algorithm 1, for $M_z^\kappa(\alpha)$ with $\alpha \in (0, 1)$ and diffusion horizon $T > 0$, and define its continuous-time interpolation*

$$Z^\epsilon(t) := Z_{\lfloor \epsilon^{-2} t \rfloor}^\epsilon, \qquad t \geq 0.$$

*Let $R_\kappa^z = d_{M_z^\kappa}(z, (W_z^\kappa)^c)$, and suppose $L \geq 1$ satisfies*

$$\sup_{x \in M_z^\kappa(\alpha)} \text{Ric}_{M_z^\kappa}(x) \geq -L^2.$$

*Then for $T < \frac{(R_z^\kappa)^2}{4k_z^\kappa L}$, as $\epsilon \to 0$, the process $Z^\epsilon$ converges in distribution to Riemannian Brownian motion stopped at the boundary of $M_z^\kappa(\alpha)$, with respect to the Skorokhod topology, on a set $C_{\kappa,z}^T \subset \Omega$ such that*

$$\mathbb{P}(C_{\kappa,z}^T) \geq 1 - \exp\left(-\frac{(R_z^\kappa)^2}{32T}\right).$$

For the proof, see Appendix D. Intuitively, in the small–step limit, our algorithm converges to Riemannian Brownian motion on $M_z^\kappa(\alpha)$, stopped at the boundary, with the deviation probability decaying exponentially in the inverse time horizon. Theorem 1 extends the main convergence result of Schwarz et al. (2022) to possibly non-compact manifolds. Importantly, Schwarz et al. (2022)

showed that achieving this convergence requires a *second-order correction* term (capturing the effect of Christoffel symbols), rather than the commonly used naive first-order update. This motivates our use of the second-order scheme in Algorithm 1. In Appendix E (Algorithm 4), we describe SORBES-SE (*Stable/Efficient*), an implementation of SORBES that approximates the second-order correction using finite differences. It employs an adaptive step size $\epsilon$ that adjusts to the local curvature of the space, while still preserving the convergence guarantees.

### 2.4 MUTANG - MUTATION ENUMERATION IN TANGENT SPACE

To further exploit the manifold structure of $M_z^\kappa$, modelling the neighborhood of a point $z \in \mathcal{Z}$, let us observe that the ambient tangent space $T_{\hat{\text{Dec}}(z)}\hat{\text{Dec}}(W_z^\kappa)$ identifies directions in peptide space along which the decoder output is the most sensitive. We interpret these directions as defining a *mutation space* for the decoded peptide $\text{p}(z)$, providing candidate amino-acid substitutions.

Formally, let $U^\kappa(z)$ (see Equation 8) denote an orthonormal basis of $T_{\hat{\text{Dec}}(z)}\hat{\text{Dec}}(W_z^\kappa)$ in the ambient space (Figure 2A–B), and let $u_j \in \mathbb{R}^{LA}$ be the $j$-th basis vector. We reshape $u_j$ into matrix form

$$\Delta \text{Dec}^{(j)}(z) = \texttt{reshape}(u_j, (L, A)) \in \mathbb{R}^{L \times A}. \tag{3}$$

Intuitively, each entry $\Delta \text{Dec}^{(j)}(z)_{\ell,a}$ measures the first-order sensitivity of the probability assigned to amino acid $\mathcal{A}_a$ at position $\ell$, thereby suggesting a possible substitution. To extract candidate mutations, we introduce a sensitivity threshold $\theta_{\text{mut}} > 0$ and declare that

$$\left| \Delta \text{Dec}^{(j)}(z)_{\ell,a} \right| \geq \theta_{\text{mut}} \quad \Rightarrow \quad \text{add mutation } \text{p}(z)_\ell \rightarrow a, \tag{4}$$

where $\text{p}(z)_\ell$ is the current residue at position $\ell$. Applying this rule across all $j = 1, \ldots, k_z^\kappa$ yields a *mutation pool*

$$\mathcal{P} \subseteq \{1, \ldots, L\} \times \mathcal{A}.$$

To enumerate candidate peptides, for each sequence position $\ell$ we define the set of admissible residues (Figure 2C) as

$$S_\ell = \{ \mathcal{A}_a \mid (\ell, a) \in \mathcal{P} \} \cup \{\text{p}(z)_\ell\},$$

i.e., all suggested mutations together with the identity residue. The complete candidate set is then obtained as the Cartesian product (Figure 2D):

$$\mathcal{C}(\text{p}(z)) = \prod_{\ell=1}^{L} S_\ell = \{ y \in \Sigma^L : y_\ell \in S_\ell \; \forall \ell \}. \tag{5}$$

The details of MUTANG are provided in Appendix F (Algorithm 5).

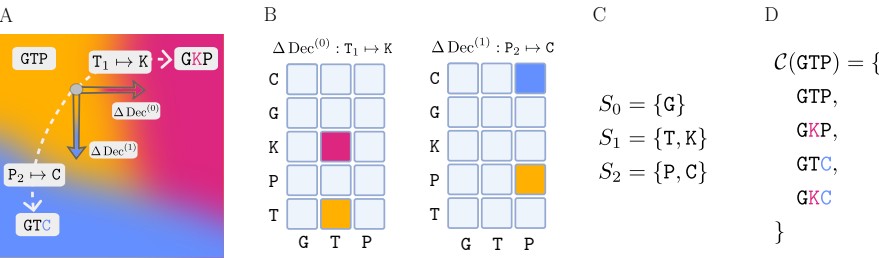

Figure 2: **Tangent space as mutation space and local enumeration. A)** Around an example peptide GTP we consider two orthogonal peptide-space tangent directions $\Delta \text{Dec}^{(j)}$ obtained from the SVD of the decoder Jacobian at the peptide code. Each direction suggests a specific substitution: $\texttt{T}_1 \rightarrow \texttt{K}$ and $\texttt{P}_2 \rightarrow \texttt{C}$. **B)** Each $\Delta \text{Dec}^{(j)}$ is reshaped into an $L \times A$ map (rows: amino acids; columns: positions). **C)** Thresholded entries define per-position sets of admissible residues (identity always included). **D)** The candidate set is the Cartesian product $\mathcal{C}(\texttt{GTP}) = \prod_\ell S_\ell$.

## 2.5 LOCAL ENUMERATION

We aim to densely populate the neighbourhood of a single prototype peptide with valid, diverse candidates. Starting from a seed peptide p with latent code $z_0$, we launch multiple Riemannian random walk trajectories using the SORBES algorithm (Sec. 2.3). At the start and after each step, the current latent state is decoded into a peptide and augmented with additional variants generated by MUTANG (Sec. 2.4). Additionally, to account for local dimension variability, we re-estimate the $\kappa$-stable submanifold $M_z^\kappa$ at each step of the random walk. The union of all decoded walk steps and tangent-space mutations yields a compact, high-quality *local candidate set* around p. LOCALENU-MERATION is presented in Algorithm 2.

---

**Algorithm 2** LOCALENUMERATION

---

**Require:** seed peptide p, $\kappa_{\text{SORBES}}, \kappa_{\text{MUTANG}} \geq 0$, number of trajectories $M$, walk time budget $T_{\text{walk}}$, nominal step size $\epsilon$, mutation threshold $\theta_{\text{mut}}$
1: Encode p to latent $z_0$;    $\mathcal{C} \leftarrow \{\text{p}\}$
2: **for** $m = 1$ to $M$ **do**
3:     $z \leftarrow z_0$; $t \leftarrow 0$
4:     $\mathcal{C} \leftarrow \mathcal{C} \cup \text{MUTANG}(z, \kappa_{\text{MUTANG}}, \theta_{\text{mut}})$
5:     **while** $t < T_{\text{walk}}$ **do**
6:       $((\_, z), \sigma) \leftarrow \text{SORBES-SE}(z, \kappa_{\text{SORBES}}, \epsilon, \text{STEP}_{\max}=1)$    (single step of SORBES-SE)
7:       $t \leftarrow t + \sigma^2$;    $\mathcal{C} \leftarrow \mathcal{C} \cup \{\text{p}(z)\}$
8:       $\mathcal{C} \leftarrow \mathcal{C} \cup \text{MUTANG}(z, \kappa_{\text{MUTANG}}, \theta_{\text{mut}})$
9:     **end while**
10: **end for**
11: **return** $\mathcal{C}$                                                (local candidate set)

---

## 2.6 LE-BO - LOCAL ENUMERATION BAYESIAN OPTIMIZATION

Finally, we integrate our LOCALENUMERATION algorithm into a Bayesian optimization (Garnett, 2023) framework for peptide design, which we term Local Enumeration Bayesian Optimization (LE-BO). Instead of performing costly optimization of the acquisition function in the latent space, which typically relies on Euclidean-distance kernels and ignores both the latent geometry and the discrete nature of peptides, we use surrogate Gaussian process models (Seeger, 2004), defined directly in the peptide space. The acquisition function is optimized by locally enumerating peptides in the vicinity of the most promising candidates and then selecting the peptide that maximizes the acquisition value. To further encourage exploration and increase the diversity of discovered candidates, we employ the ROBOT scheme (Maus et al., 2023), which promotes searching across a broader set of promising regions. Details of LE-BO are presented in Algorithm 3.

## 2.7 POGS - POTENTIAL-MINIMIZING GEODESIC SEARCH

Given two prototype peptides with latent vectors $z_a$ and $z_b$, we aim to generate *seeds*, i.e., *analogues that are jointly similar to both vectors and have high predicted activity*. To this end, we construct a discrete geodesic-like curve connecting $z_a$ and $z_b$, interpreted physically as a system with *kinetic energy* (geometric term) and an added *potential energy* (property term). This provides a natural tradeoff between similarity to both seeds and the desired molecular property.

Because the decoder Jacobian may have varying rank, we avoid intrinsic pullback computations and work in the ambient peptide-probability space $\mathbb{R}^{LA}$. For a sequence of latent waypoints $Z = \{z_0{=}z_a, z_1, \dots, z_N{=}z_b\}$, we define their decoded logits $X_k = \log(\hat{\text{Dec}}(z_k))$, and approximate curve length using *chord distances* $\|X_{k+1} - X_k\|_2$. This extrinsic metric serves as a first-order surrogate for geodesic energy, bypassing costly Christoffel evaluations and remaining stable under rank variability.

We define the total energy of a discrete path $Z$ as

$$\mathcal{E}_{\lambda,\mu}(Z) = \underbrace{\sum_{k=0}^{N-1} \|X_{k+1} - X_k\|_2^2}_{\text{kinetic term: geometric similarity}} + \lambda \underbrace{\sum_{k=0}^{N} \Phi(X_k)}_{\text{potential term: property bias}} + \mu \underbrace{\sum_{k=0}^{N-1} \|z_{k+1} - z_k\|_2^2}_{\text{latent regularizer}}, \quad (6)$$

---

**Algorithm 3** LE-BO

---

**Require:** ORACLE function to be optimized; seed $\mathrm{p_{seed}}$; maximum budget $B_{\max}$; trust region distance $d_{\mathrm{trust}}$; number of ROBOT evaluations per iteration $k_{\mathrm{ROBOT}}$; diversity threshold $d_{\mathrm{ROBOT}}$; a surrogate Gaussian Process GP model with GP. acquistion function.

1: $\mathrm{p_{current}} := \mathrm{p}_{best} := \mathrm{p_{seed}}, \mathcal{D} := \{\mathrm{p_{seed}}\}, \mathcal{E} = \{(\mathrm{p_{seed}}, \mathrm{ORACLE}(\mathrm{p_{seed}}))\}$
2: **for** $iter := 1$ to $\lfloor B_{max}/k_{\mathrm{ROBOT}} \rfloor$ **do**
3:     $\mathcal{D} := \mathcal{D} \cup \mathrm{LOCALENUMERATION}(\mathrm{p_{current}})$            Explore the neighborhood of $\mathrm{p_{current}}$
4:     $\mathrm{GP} := \mathrm{GP.\,fit}(\mathcal{E})$                                             Fit the surrogate GP model
5:     $\mathcal{D}_{\mathrm{trust}} := \{\mathrm{p} \in \mathcal{D} \mid \mathrm{Levenshtein}(\mathrm{p}, \mathrm{p_{best}}) \leq d_{\mathrm{trust}}\}$            Define the trust region
6:     **for** $i := 1$ to $k_{\mathrm{ROBOT}}$ **do**
7:        $\mathrm{p^i_{ROBOT}} := \arg\max_{\mathrm{p} \in \mathcal{D}_{\mathrm{trust}}} \mathrm{GP.\,acquistion}(\mathrm{p})$
8:        $\mathcal{E} := \mathcal{E} \cup \{(\mathrm{p^i_{ROBOT}}, \mathrm{ORACLE}(p_{\mathrm{ROBOT}}))\}$
9:        $\mathcal{D}_{\mathrm{trust}} := \mathcal{D}_{\mathrm{trust}} \setminus \{\mathrm{p} \in \mathcal{D} \mid \mathrm{Levenshtein}(\mathrm{p}, \mathrm{p^i_{ROBOT}}) \leq d_{\mathrm{ROBOT}}\}$    ROBOT diversity filtering
10:    **end for**
11:     $p_{\mathrm{current}} := \arg\min_{1 \leq i \leq k_{\mathrm{ROBOT}}} \mathrm{ORACLE}(\mathrm{p^i_{ROBOT}})$
12:     **if** $\mathrm{ORACLE}(\mathrm{p_{current}}) \geq \mathrm{ORACLE}(\mathrm{p_{best}})$ **then**
13:        $\mathrm{p_{best}} := \mathrm{p_{current}}$
14:     **end if**
15: **end for**
16: **return** $\mathrm{p_{best}}$

---

where $\Phi$ is the property prediction (e.g. negative log MIC), $\lambda \geq 0$ balances geometry vs. property, and $\mu \geq 0$ regularizes latent jumps to discourage large chords in $\mathcal{Z}$. The first term corresponds to kinetic energy (favoring smooth, short ambient curves), the second to a potential energy that biases toward low $\Phi$, and the third acts as a stabilizer ensuring robustness of the chord points in the latent space.

To perform optimization and search, we initialize $Z$ by straight-line interpolation in latent space, and later optimize $\mathcal{E}_{\lambda,\mu}(Z)$ w.r.t. $Z$ using ADAM solver. During optimization, only the interior points $z_1, \ldots, z_{N-1}$ are updated. Given an optimized path $Z$, we decode each $z_i$ to a peptide $\mathrm{p}_i$ and remove consecutive duplicates, obtaining a *peptide path* $(\mathrm{p}'_0, \ldots, \mathrm{p}'_{N'})$ of length $N'$. We call a peptide $\mathrm{p}'_k$ a *seed* if its potential satisfies $\Phi(\mathrm{p}'_k) \leq \theta_{\mathrm{pot}}$ for a threshold $\theta_{\mathrm{pot}}$. A peptide $\mathrm{p}'_k$ is called a *well* if it is a seed and also a local minimum of the potential $\Phi$ along the peptide path. The detailed algorithm is presented in Appendix G (Algoritm 6).

# 3 RESULTS

## 3.1 PoGS EVALUATION

To evaluate the PoGS procedure, we applied it for the HydrAMP model (Szymczak et al., 2023), using average standardized MIC predictions against 3 *Escherichia coli* strains (*E. coli* ATCC11775, AIG221 and AIG222) of a APEX-derived transformer prediction (Wan et al., 2024) (Appendix H) as the potential function. 300 prototype pairs $(z_a, z_b)$ were drawn from the Veltri dataset (Veltri et al., 2018), restricted to active peptides (average MIC $\leq 32\,\mu\mathrm{g/ml}$ against 3 *E. coli* strains) with edit distance between prototypes $\geq 10$.

For each pair $z_a$ and $z_b$, we compared three paths between $z_a$ and $z_b$: straight Euclidean interpolation, PoGS without potential, and full PoGS ; Sec. 2.7), measuring chord latent and ambient lengths, decoded peptide path lengths, and property-based counts of seeds and wells. PoGS hyperparametrs and metrics are described in Appendix G. PoGS achieved shorter ambient paths, and substantially more seeds and wells (Table 1), showing that property-aware potentials enrich trajectories for active candidates.

## 3.2 LE-BO EVALUATION

We next evaluated our LE-BO algorithm on a black-box peptide optimization task with a budget of 1400 evaluations. Optimization was initialized from four seed peptides derived using PoGS

Table 1: **PoGS results.** Comparison of straight interpolation, geodesic (no potential), and property-extended geodesic. Reported are latent length, ambient length, peptide path per length, and counts of seeds and wells.

| Method | Latent length | Ambient length | Peptide path length | Potential | Seeds | Wells |
|---|---|---|---|---|---|---|
| Straight interpolation | **6.43 ± 0.13** | 1601.56 ± 57.04 | 71.25 ± 6.12 | -501.00 ± 23.92 | 11.70 ± 5.54 | 2.50 ± 1.90 |
| PoGS w.o. potential ($\lambda = 0$) | 7.98 ± 0.18 | **1240.03 ± 40.72** | **66.68 ± 4.23** | -603.34 ± 15.12 | 18.90 ± 8.40 | 6.40 ± 3.20 |
| PoGS ($\lambda = 0.01$) | 9.02 ± 0.09 | 1432.55 ± 19.98 | 77.12 ± 6.12 | **-785.45 ± 34.12** | **22.70 ± 10.49** | **12.60 ± 4.45** |

for $\lambda = 0.01$ and $\mu = 0.1$ (Sec. 2.7): *KY14*, *KF16*, *KK16*, *FL14*, as well as two previously described AMPs (*mammuthusin-3*, *hydrodamin-2*) (Wan et al., 2024). The optimization procedure was repeated 10 times, and for each run the best value achieved across all optimization steps was recorded and reported in Table 3.3. LE-BO hyperparameters are described in Appendix I.

We compared LE-BO against a diverse set of state-of-the-art methods, including generative models: HydrAMP (Szymczak et al., 2023) with different creativity hyperparameter $\tau$ and PepCVAE (Das et al., 2018); the diffusion-based model LaMBO-2 (Gruver et al., 2023); evolutionary strategies: Latent CMA-ES and Relaxed CMA-ES (Hansen, 2016), where the former operates in the HydrAMP latent space and the latter optimizes directly on a continuous relaxation of one-hot sequence encodings, AdaLead (Sinai et al., 2020), PEX (Ren et al., 2022); GFlowNet-based approaches: GFN-AL (Jain et al., 2022) and GFN-AL-$\delta$CS (Kim et al., 2025); the reinforcement learning method DyNA-PPO (Angermueller et al., 2020); probabilistic methods: CbAS (Brookes et al., 2019) and Evolutionary BO (Sinai et al., 2020); the insertion-based Joker method (Porto et al., 2018); and greedy Random Mutation (González-Duque et al., 2024). Comparison to competitor latent BO methods SAASBO (Eriksson & Jankowiak, 2021), Hvarfner's Vanilla BO (Hvarfner et al., 2024), and LineBO (Kirschner et al., 2019) was not feasible due to their relative computational inefficiency: while LE-BO finalized within 2 hours, for all those methods, a single run of the same number of 1400 steps within HydrAMP latent space did not finish within a week.

Additionally, we conducted an ablation study on LE-BO variants to isolate the effects of random walks and mutation enumeration. These variants modify the ENUMERATELOCAL sub-procedure in Algorithm 2. The *Euclidean walk* variant replaces SORBES with a naive Euclidean random walk in the latent space. The *mutation-disabled* variant omits MUTANG. Finally, the *walk-disabled* variant uses only a single MUTANG without random walks.

As the optimized black-box function, we used the APEX MIC regressor (Wan et al., 2024), with the objective of minimization. We minimize the average $\log_2$ MIC across three *E. coli* strains, reporting the mean over the best results from 10 repeated optimization runs (Table 3.3).

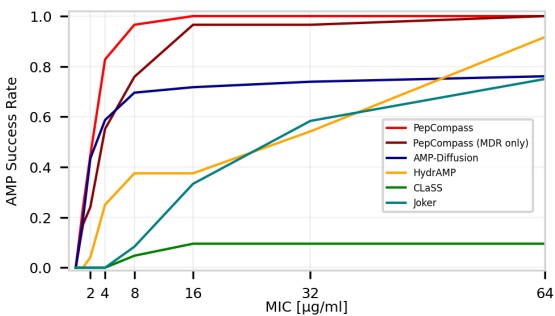

Figure 3: **Antimicrobial peptide success rates across MIC thresholds**. Success rate is defined as the fraction of generated peptides with MIC below the specified threshold against at least one tested strain. Results are based on experimental validation against 19 bacterial strains, including 8 MDR isolates.

LE-BO achieved the best performance on five out of six prototypes, outperforming all baselines except for *FL14*, where it ranked second. The Ablated LE-BO variant with Euclidean walk and enabled mutations achieved the second-best performance on four out of six prototypes, underscoring the importance of mutation enumeration in the optimization process. These ablations further confirmed that both Riemannian random walks and mutations are essential for consistently achieving low MIC values.

Table 2: Minimal $\log_2$ MIC values achieved by each optimization method. Reported values are the mean and standard deviation over 10 runs. The top row shows the predicted $\log_2$ MIC of seeds before optimization. The first block reports baseline results, the second block shows ablations, and the last row presents our method. **Bold:** best overall value for a prototype. Underline: second-best.

| Method | | | KY14 | KF16 | KK16 | FL14 | mammuthusin-3 | hydrodamin-2 |
|---|---|---|---|---|---|---|---|---|
| $\log_2$ MIC value (seed) | | | 2.88 | 3.39 | 2.71 | 4.00 | 4.30 | 6.96 |
| GFN-AL | | | $2.73 \pm 0.27$ | $3.20 \pm 0.19$ | $2.63 \pm 0.17$ | $3.73 \pm 0.47$ | $4.30 \pm 0.00$ | $3.85 \pm 0.76$ |
| GFN-AL-$\delta$CS | | | $1.91 \pm 0.19$ | $1.74 \pm 0.20$ | $1.79 \pm 0.08$ | $2.06 \pm 0.42$ | $3.02 \pm 0.19$ | $1.77 \pm 0.41$ |
| PEX | | | $1.39 \pm 0.07$ | $1.48 \pm 0.09$ | $1.54 \pm 0.14$ | $1.27 \pm 0.27$ | $2.65 \pm 0.09$ | $1.14 \pm 0.11$ |
| Joker | | | $4.66 \pm 2.04$ | $3.98 \pm 1.52$ | $3.79 \pm 1.61$ | $4.22 \pm 1.08$ | $6.49 \pm 1.75$ | $6.28 \pm 0.33$ |
| Random Mutation | | | $1.23 \pm 0.26$ | $1.17 \pm 0.13$ | $0.97 \pm 0.40$ | $1.02 \pm 0.59$ | $2.12 \pm 0.80$ | $0.69 \pm 0.19$ |
| LaMBO-2 | | | $1.88 \pm 0.25$ | $2.11 \pm 0.20$ | $1.72 \pm 0.18$ | $1.76 \pm 0.32$ | $2.75 \pm 0.15$ | $2.17 \pm 0.35$ |
| Relaxed CMA-ES | | | $1.92 \pm 0.13$ | $1.83 \pm 0.28$ | $1.87 \pm 0.30$ | $1.93 \pm 0.40$ | $2.86 \pm 0.29$ | $1.66 \pm 0.34$ |
| Latent CMA-ES | | | $1.81 \pm 0.33$ | $2.07 \pm 0.56$ | $1.72 \pm 0.29$ | $1.72 \pm 0.20$ | $2.17 \pm 0.60$ | $1.87 \pm 0.44$ |
| CbAS | | | $2.88 \pm 0.00$ | $3.39 \pm 0.00$ | $2.71 \pm 0.00$ | $4.00 \pm 0.00$ | $4.30 \pm 0.00$ | $5.50 \pm 0.93$ |
| DyNAPPO | | | $1.45 \pm 0.36$ | $1.31 \pm 0.22$ | $1.34 \pm 0.20$ | $0.73 \pm 0.53$ | $2.42 \pm 0.54$ | $0.82 \pm 0.39$ |
| Evolutionary BO | | | $1.65 \pm 0.23$ | $1.45 \pm 0.42$ | $1.50 \pm 0.12$ | $1.70 \pm 0.19$ | $2.68 \pm 0.12$ | $1.28 \pm 0.33$ |
| AdaLead | | | $0.87 \pm 0.49$ | $1.01 \pm 0.28$ | $0.93 \pm 0.24$ | $\mathbf{0.51 \pm 0.38}$ | $2.30 \pm 0.28$ | $0.66 \pm 0.21$ |
| PepCVAE | | | $3.66 \pm 0.00$ | $2.87 \pm 0.01$ | $3.14 \pm 0.11$ | $2.12 \pm 0.00$ | $4.30 \pm 0.00$ | $6.74 \pm 0.06$ |
| HydrAMP $\tau = 5.0$ | | | $2.35 \pm 0.08$ | $2.03 \pm 0.12$ | $1.88 \pm 0.10$ | $2.19 \pm 0.12$ | $3.19 \pm 0.27$ | $2.39 \pm 0.38$ |
| HydrAMP $\tau = 2.0$ | | | $2.60 \pm 0.03$ | $2.27 \pm 0.02$ | $2.11 \pm 0.11$ | $2.81 \pm 0.30$ | $3.99 \pm 0.01$ | $5.02 \pm 0.28$ |
| HydrAMP $\tau = 1.0$ | | | $2.86 \pm 0.01$ | $2.27 \pm 0.00$ | $2.35 \pm 0.00$ | $3.72 \pm 0.35$ | $4.27 \pm 0.10$ | $6.09 \pm 0.01$ |
| | Walk | Mutation | | | | | | |
| | Euclidean | – | $1.37 \pm 0.27$ | $1.33 \pm 0.23$ | $1.24 \pm 0.18$ | $1.29 \pm 0.17$ | $0.91 \pm 0.37$ | $1.16 \pm 0.24$ |
| | SORBES-SE | – | $1.37 \pm 0.22$ | $1.42 \pm 0.22$ | $1.12 \pm 0.35$ | $1.18 \pm 0.20$ | $1.07 \pm 0.50$ | $1.12 \pm 0.13$ |
| Ablated LE-BO | – | ✓ | $1.71 \pm 0.20$ | $1.46 \pm 0.40$ | $1.82 \pm 0.13$ | $1.24 \pm 0.13$ | $1.90 \pm 0.42$ | $1.12 \pm 0.20$ |
| | Euclidean | ✓ | $0.65 \pm 0.18$ | $0.71 \pm 0.18$ | $0.83 \pm 0.32$ | $0.87 \pm 0.22$ | $0.78 \pm 0.45$ | $0.80 \pm 0.18$ |
| LE-BO | SORBES-SE | ✓ | $\mathbf{0.50 \pm 0.24}$ | $\mathbf{0.60 \pm 0.29}$ | $\mathbf{0.50 \pm 0.14}$ | $0.60 \pm 0.22$ | $\mathbf{0.50 \pm 0.38}$ | $\mathbf{0.58 \pm 0.34}$ |

## 3.3 WET-LAB VALIDATION

To validate the computational predictions from PoGS and LE-BO, we conducted comprehensive *in vitro* antimicrobial testing of the generated peptides. A total of 29 novel peptides were experimentally evaluated: 4 seed peptides discovered through PoGS bi-prototype geodesics with property-aware potentials, and 25 analogs derived from these 4 seeds through LE-BO optimization for *E. coli* activity. These peptides were tested against a panel of 19 bacterial strains, including 8 multidrug-resistant (MDR) isolates (Appendix J), to assess both broad-spectrum activity and efficacy against clinically relevant resistant pathogens. We compared the success rate of PepCompass to previous methods that were also validated experimentally (HydrAMP (Szymczak et al., 2023), AMP-Diffusion (Torres et al., 2025), CLaSS (Das et al., 2020), Joker (Porto et al., 2018)). To this end, for each activity threshold, we computed the fraction of tested peptides that were active against at least one bacterial strain with this activity thershold.

As demonstrated in Figure 3, with unprecedented $100\%$ success rate for the standard activity threshold of $32\mu$g/ml and $82\%$ rate at much more demanding threshold of $4\mu$g/ml, PepCompass achieved superior performance across various MIC thresholds compared to previous methods, maintaining high success rates even when evaluated specifically against MDR strains. The experimental results confirmed the expected activity increase from prototypes to seeds to analogs for Gram-negative bacteria, directly validating our optimization strategy that targeted *E. coli* activity. Indeed, while the generated peptides showed some activity against Gram-positive bacteria (Figure 8), the clear enhancement from optimization was primarily observed against Gram-negative pathogens (Figure 9).

## CONCLUSIONS

By leveraging Riemannian latent geometry, interpretable tangent-space mutations, and potential-augmented geodesics, PepCompass enables efficient navigation of and optimization within peptide space across global and local scales. One of possible limitation of our approach is the reliance on the Eucleadian-distance based metric on decoder outputs in the ambient space. However, any other metrics that would better capture the ambient manifold could easily be incorporated. Already now, our computational experiments demonstrate superior performance over state-of-the-art baselines, and wet-lab validation confirms unprecedented success rates, with all tested peptides showing activity *in vitro*, including activity against multidrug-resistant pathogens. These results establish geometry-

aware exploration as a powerful new paradigm for controlled generative design in vast biological spaces.

REPRODUCIBILITY STATEMENT

All proofs, together with their explanations and underlying assumptions, are provided in Appendices B, C and D. All implementation details and hyperparameters used in the experiments are listed in Appendices E, F, G, H and I. Additionally, we release the source code and experimental configurations necessary to reproduce the key results. Full details of wet-lab validation procedure are described in Appendix J.

ETHICS STATEMENT

This work includes methods generally applicable to peptide design. An example of potential malicious use of PepCompass would include optimization of peptide toxicity. However, the intention of this paper is to instead provide tools facilitating the design of therapeutic peptides.

LLM USAGE

Large Language Models (LLMs) were used in this work to improve the clarity and structure of the text. Their use was limited to rephrasing and stylistic refinement. In addition, LLMs were employed to support the search of related work, helping to verify the accuracy of claims about prior research.

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

# APPENDIX

## A  RELATED WORK

**Riemannian latent geometry.** Deep generative models can be endowed with Riemannian structure (do Carmo, 1992) by pulling back the ambient metric through the decoder Jacobian (Arvanitidis et al., 2018; 2020; Shao et al., 2018). This allows distances and geodesics to reflect data geometry rather than Euclidean latent coordinates. However, most approaches assume a single smooth manifold of fixed dimension. Evidence from both theory and experiments shows that data often lie on unions of manifolds with varying intrinsic dimension or CW-complex structures (Lou, 2023; Brown et al., 2022; Wang & Wang, 2024). In practice, existing methods either restrict the latent to very low dimensions ($\leq 8$) or inflate the metric with variance terms to ensure full rank (Arvanitidis et al., 2020; Detlefsen et al., 2022), but these ignore extrinsic geometry. Our model instead *decomposes the latent into Riemannian submanifolds of varying dimensions*, enabling principled geometry across heterogeneous regions.

**Brownian motion and random walks.** Latent Brownian motion has been used as a prior for VAEs (Kalatzis et al., 2020) and for Riemannian score-based modeling (De Bortoli et al., 2022). But these approaches rely on first-order updates. Convergence results for geodesic random walks show that correct Riemannian and sub-Riemannian Brownian motion requires second-order approximations (Schwarz et al., 2022; Herrmann et al., 2023). Our method explicitly incorporates this requirement, yielding diffusion-consistent walks where previous methods diverge.

**Tangent spaces and interpretability.** Tangent-space analysis has mostly been applied in vision, where interpretable latent directions are discovered in GANs or diffusion models via Jacobian or eigen decompositions (Shen et al., 2020; Park et al., 2021; Wang et al., 2024; Alemi et al., 2023). Frames induced by augmentations provide another lens on local tangent geometry (Schneider et al., 2022). We are the first to provide an *interpretable tangent space in peptide sequence models*, where tangent vectors correspond directly to biologically meaningful mutations.

**Geodesics and potentials.** Geodesics are widely used for interpolation and counterfactual reasoning in latent space (Pegios et al., 2024; Blondel et al., 2024). Yet these are typically free geodesics. We extend the concept with *potentials*, leveraging the Jacobi metric (Gibbons, 2015) so that peptide traversals account for both geometry and biochemical preferences.

**Applications in molecules and proteins.** Geometry-aware latent models have been used for chemical-space exploration (Zhong et al., 2022; Winter et al., 2022), molecule optimization (Feng et al., 2021), and protein sequence modeling (Cao et al., 2022; Detlefsen et al., 2022). Our approach complements these by combining: (i) varying-dimension latent decomposition, (ii) second-order consistent Brownian walks, (iii) interpretable tangent spaces via mutations, and (iv) potential-augmented geodesics tailored to peptide design.

## B $\kappa$-STABLE DIMENSION OF THE PEPCVAE (DAS ET AL., 2018) AND HYDRAMP SZYMCZAK ET AL. (2023) MODELS

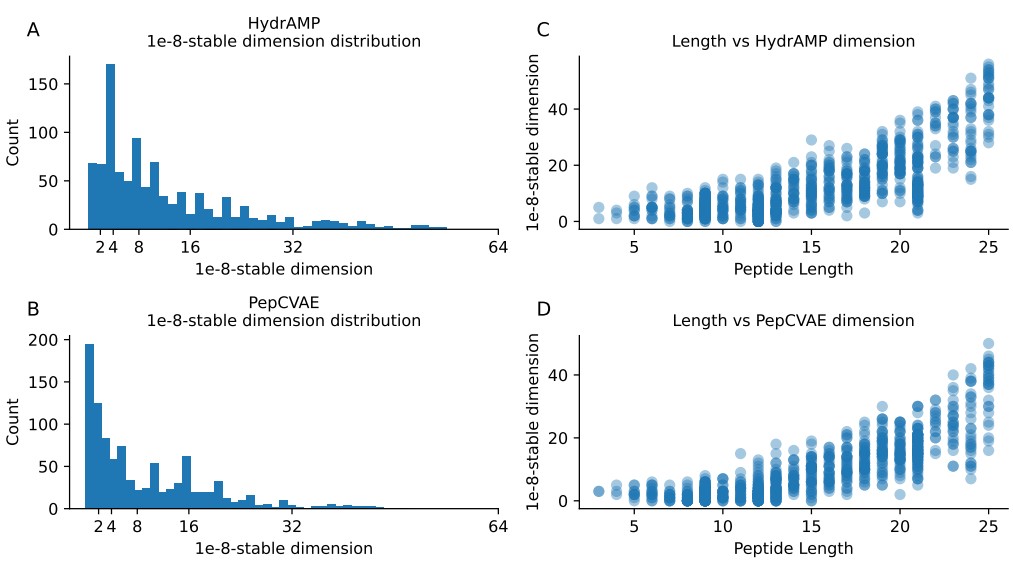

Figure 4: **Stable rank and peptide statistics. A–B)** Distributions of the $\kappa$-stable dimension ($\kappa = 10^{-8}$) for HydrAMP (A) and PepCVAE (B) across sampled peptides. **C–D)** Scatter plots of peptide length versus $\kappa$-stable dimension for HydrAMP (C) and PepCVAE (D), revealing a clear positive correlation: longer peptides tend to yield higher stable dimensions.

To quantify how the $\kappa$-stable dimension varies across the latent space, we sampled 1,000 points from the HydrAMP (Szymczak et al., 2023) training set and computed their $\kappa$-stable dimensions under both the PepCVAE (Das et al., 2018) and HydrAMP (Szymczak et al., 2023) models. We set $\kappa = 10^{-8}$, corresponding to the precision of the `float32` format commonly used in neural network computations. This choice ensures that no eigenvalue of the inverse metric tensor exceeds $10^8$, thereby avoiding numerical instabilities.

As shown in Figure 4A, the $\kappa$-stable dimension was always strictly below the latent dimensionality (64) of both models. This indicates that the effective local dimensionalities of the peptide spaces are substantially smaller than the nominal latent dimension. Furthermore, when comparing peptide length (Figure 4B) to $\kappa$-stable dimension (Figure 4C), we observe a clear positive correlation: longer peptides systematically yield higher $\kappa$-stable dimensions. Intuitively, this suggests that longer sequences admit more locally meaningful perturbations, which naturally translate into a richer set of candidate substitutions. This observation further justifies our MUTANG strategy (§2.4), as it allocates a larger and more diverse mutation pool precisely where biological sequence length provides greater combinatorial flexibility.

## C    CONSTRUCTION OF $\kappa$-STABLE MANIFOLDS

In this section we will introduce a construction of $\kappa$-stable Riemannian submanifolds, namely

$$\mathbb{M}^\kappa = \{M_z^\kappa : z \in \mathcal{Z}\}, \qquad M_z^\kappa = (W_z^\kappa, G_{\text{Dec}}),$$

where each $W_z^\kappa \ni z$ is an open affine submanifold of *maximal dimension* (denoted $k_z^\kappa$) through $z$, such that the pullback metric $G_{\text{Dec}}$ restricted to $W_z^\kappa$, denoted $G_{\text{Dec}}|_{W_z^\kappa}$, has full rank and satisfies the $\kappa$-stability condition

$$\inf_{\substack{v \in T_z W_z^\kappa \\ \langle v,v \rangle_{\mathbb{R}^d} = 1}} \langle v, v \rangle_z^{\text{Dec}} > \kappa^2.$$

For this, we will use the truncated-SVD of a flattened decoder Jacobian $J_{\hat{Dec}}$. Let

$$J_{\hat{\text{Dec}}}(z) = U(z)\,\Sigma(z)\,V(z)^\top$$

be the thin SVD of the decoder Jacobian, with singular values $\sigma_0(z) \geq \sigma_1(z) \geq \cdots \geq 0$, and $\Sigma(z) = \text{diag}\,((\sigma_0, \sigma_1, \ldots, \sigma_{d-1}))$. Now let us note that:

$$G_{\text{Dec}}(z) = J_{\hat{\text{Dec}}}(z)^\top J_{\hat{\text{Dec}}}(z) = V(z)\Sigma^2(z)V(z)^\top, \tag{7}$$

and define the $\kappa$-*stable dimension*:

$$k_z^\kappa = \#\{\, i : \sigma_i(z)^2 > \kappa \,\}.$$

Truncating SVD-decomposition to first $k_z^\kappa$ singular values gives

$$U^\kappa(z) \in \mathbb{R}^{(LA) \times k_z^\kappa}, \quad \Sigma_z^\kappa \in \mathbb{R}^{k_z^\kappa \times k_z^\kappa}, \quad V^\kappa(z) \in \mathbb{R}^{d \times k_z^\kappa}, \tag{8}$$

with truncated Jacobian:

$$J_{\hat{\text{Dec}}}^\kappa = U(z)^\kappa \Sigma(z)^\kappa V(z)^\kappa \in \mathbb{R}^{LA \times d}.$$

We then define the affine subspace (together with its parametrization)

$$\mathcal{V}_z^\kappa = \{\phi_z^\kappa(x) : x \in \mathbb{R}^{k_z^\kappa}\}, \qquad \phi_z^\kappa(x) = z + V^\kappa(z)x.$$

Restricting the decoder to $\mathcal{V}_z^\kappa$ gives $\text{Dec}_z^\kappa = \hat{\text{Dec}} \circ \phi_z^\kappa$, with Jacobian

$$J_{\text{Dec}_z^\kappa}(0) = U^\kappa(z)\Sigma^\kappa(z), \tag{9}$$

which has full column rank $k_z^\kappa$. By the inverse function theorem, there exists a neighborhood $\hat{W}_z^\kappa = B(0, r_z^\kappa)$ such that $\text{Dec}_z^\kappa(\hat{W}_z^\kappa)$ is a smooth $k_z^\kappa$-dimensional manifold. Its pullback metric is

$$G_{\text{Dec}_z^\kappa}(0) = \left(\Sigma^\kappa(z)\right)^2, \tag{10}$$

with eigenvalues $\{\sigma_i(z)^2 : \sigma_i(z)^2 > \kappa\}$, all $\geq \kappa$.

Finally, set $W_z^\kappa = \phi_z^\kappa(\hat{W}_z^\kappa)$. Then

$$M_z^\kappa = (W_z^\kappa, G_{\text{Dec}}),$$

and $(\phi_z^\kappa)^{-1}$ is a diffeomorphic isometry between $(W_z^\kappa, G_{\text{Dec}})$ and $(\hat{W}_z^\kappa, G_{\text{Dec}_z^\kappa})$. From this and Eq. 10 it follows that $M_\kappa^z$ satisfies the $\kappa$-stability condition.

The maximality of $k_z^\kappa$ follows from the fact that if there existed an affine subspace $\bar{W}_\kappa^z$ with $\dim(\bar{W}_\kappa^z) > k_z^\kappa$ such that $G_{\text{Dec}}|_{\bar{W}_z^\kappa}$ has full rank and satisfies the $\kappa$-stability condition, then we could define

$$\bar{W}_z^\kappa \cap (W_z^\kappa)^\perp = \left\{ w \in \bar{W}_\kappa^z : \forall v \in W_z^\kappa, \ \langle v, w \rangle^{\text{Euc}} = 0 \right\} \subset \bar{W}_z^\kappa,$$

as the subspace of $\bar{W}_z^\kappa$ orthogonal to $W_z^\kappa$. Since $\dim(\bar{W}_\kappa^z) > \dim(W_\kappa^z)$, it follows that $\dim\!\left(\bar{W}_z^\kappa \cap (W_z^\kappa)^\perp\right) > 0$. By construction

$$\bar{W}_z^\kappa \cap (W_z^\kappa)^\perp \subset \text{span}\{V(z)_{:,k_z^\kappa}, \ldots, V(z)_{:,d}\}.$$

what implies that for all $v \in \bar{W}_z^\kappa \cap (W_z^\kappa)^\perp$ it holds that

$$v = a_{k_z^\kappa} V(z)_{:,k_z^\kappa} + \cdots + a_{d-1} V(z)_{:,d-1},$$

for some $a_{k_z^\kappa}, \ldots, a_{d-1} \in \mathbb{R}$. Now take $v \in \bar{W}_z^\kappa \cap (W_z^\kappa)^\perp$ such that $\langle v, v \rangle^{\text{Euc}} = 1$. Equation 7 then implies

$$\langle v, v \rangle_z^{\text{Dec}} = a_{k_z^\kappa}^2 \sigma_{k_z^\kappa}^2 \| V(z)_{:,k_z^\kappa} \|^2 + \cdots + a_{d-1}^2 \sigma_{d-1}^2 \| V(z)_{:,d} \|^2 \qquad (11)$$

$$\leq \kappa \left( a_{k_z^\kappa}^2 \| V(z)_{:,k_z^\kappa} \|^2 + \cdots + a_{d-1}^2 \| V(z)_{:,d-1} \|^2 \right) \qquad (12)$$

$$= \kappa, \qquad (13)$$

which contradicts the assumption that $\bar{V}_z^\kappa$ satisfies the $\kappa$-stability condition.

**Note.** This construction cannot be replaced by a direct Frobenius theorem argument, since $k_z^\kappa = \ell$ at a point does not imply that $k^\kappa$ is constant in a neighborhood (see Appendix C.1).

### C.1   ON THE LOCAL INSTABILITY OF $\kappa$-STABLE DIMENSION

Recall that $k_z^\kappa = \#\{ i : \sigma_i(z) > \sqrt{\kappa} \}$ counts the number of singular values of $J_{\text{Dec}}(z)$ exceeding the threshold $\sqrt{\kappa}$. While $k_z^\kappa$ is well defined at every point $z$, it need not be locally constant. In particular, singular values of $J_{\text{Dec}}(z)$ depend continuously on $z$, but they can cross the threshold $\sqrt{\kappa}$ arbitrarily close to a given point. Hence, even if $k_z^\kappa = \ell$ at some $z$, there may exist nearby points $z'$ with $k_{z'}^\kappa > \ell$ (see Figure 5).

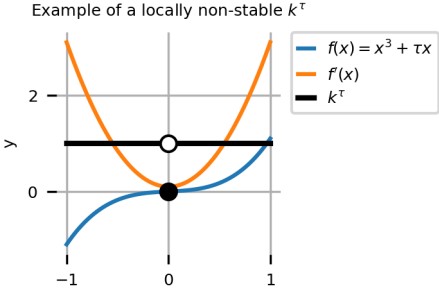

Figure 5: **Example of a local non-stability of a $\kappa$-stable dimension.** A function $f(x) = x^3 + \kappa x$ has a stable rank $k^\kappa$ equal to 1 everywhere except of 0. So in every neighbourhood of 0, a stable rank is different than 0, thus preventing the application of a Frobenious theorem.

This observation prevents a direct application of the Frobenius or constant rank theorem, which require a rank function that is constant in a neighborhood. Our construction in Sec. 2.1 circumvents this issue by working with an open set $W_z^\kappa$ around $z$ on which the rank remains constant, thereby ensuring that both $M_z^\kappa$ and $\text{Dec}(W_z^\kappa)$ are smooth $k_z^\kappa$-dimensional submanifolds.

## D   PROOF OF THEOREM 1

In this section we prove the main theorem of the paper. As preparation, we first recall the definition of Riemannian Brownian motion, starting with the compact case.

**Riemannian Brownian Motion (compact case).**   Let $(M, g)$ be a smooth, compact, connected, $d$-dimensional Riemannian manifold with Riemannian metric $g$. A *Riemannian Brownian motion* on $M$ is a continuous stochastic process

$$B = \{B_t\}_{t \geq 0}$$

defined on a filtered probability space $(\Omega, \mathcal{F}, \{\mathcal{F}_t\}_{t \geq 0}, \mathbb{P})$, satisfying:

1. $B_0 = x \in M'$ almost surely, for some fixed starting point $x \in M$.

2. The sample paths $t \mapsto B_t$ are almost surely continuous and adapted to the filtration $\{\mathcal{F}_t\}$.

3. For every smooth function $f \in C^\infty(M)$, the process

$$f(B_t) - f(B_0) - \tfrac{1}{2} \int_0^t (\Delta_g f)(B_s)\, ds$$

   is a real-valued local martingale, where $\Delta_g$ denotes the Laplace–Beltrami operator associated with $g$.

4. The generator of $B_t$ is $\frac{1}{2}\Delta_g$, i.e.

$$\lim_{t \to 0} \frac{\mathbb{E}[f(B_t)] - f(x)}{t} = \tfrac{1}{2}(\Delta_g f)(x), \quad \forall f \in C^\infty(M).$$

**Extension to the non-compact case: smooth spherical-cap compactification.** Our manifolds of interest, $M_z^\kappa(\alpha)$, $\alpha \in (0,1)$, $z \in \mathcal{Z}$, are open subsets of $M_z^\kappa$ and therefore non-compact. To define Brownian motion in this setting, one needs to control the behaviour of paths near the boundary. Classical approaches include: (i) compactification (Wang, 2010), (ii) stopping the process at the boundary (Hsu, 2002b), or (iii) reflecting it (Du & Hsu, 2021). In our work we adopt a compactification strategy via a smooth spherical cap (see Figure 6), followed by stopping on the boundary of a natural embedding of $M_z^\kappa(\alpha)$.

Concretely, let $k_z^\kappa$ be the $\kappa$-stable dimension and consider the unit sphere

$$S_1^{k_z^\kappa} \subset \mathbb{R}^{k_z^\kappa + 1}.$$

Take an atlas of this sphere consisting of two charts $(U_1, \psi_1)$, $(U_2, \psi_2)$ such that $U_1, U_2 \subset \mathbb{R}^{k_z^\kappa}$ $\psi_1(U_1) \cup \psi_2(U_2) = S_1^{k_z^\kappa}$, with $\alpha \bar{W}_z^\kappa \subset U_1$ ($\alpha \bar{W}_z^\kappa = \{\alpha v : v \in \bar{W}_z^\kappa\}$) and

$$\psi_1 \circ \psi_2^{-1}(U_2) \cap \bar{W}_z^\kappa = \emptyset.$$

Let $G_{\psi_i}$ be the pullback metric on $U_i$ induced by $\psi_i$.

Choose a smooth bump function $b \in C^\infty$ such that

$$b \equiv 1 \quad \text{on } (\tfrac{2}{3}\alpha + \tfrac{1}{3})\bar{W}_z^\kappa, \qquad b \equiv 0 \quad \text{on } (\tfrac{1}{3}\alpha + \tfrac{2}{3})\bar{W}_z^\kappa.$$

We then define the compactified manifold $\overline{M_z^\kappa}(\alpha)$ with charts $\{(U_1, \psi_1), (U_2, \psi_2)\}$ and Riemannian metric on $U_1$ given by

$$G = b\, G_{\mathrm{Dec}_z^\kappa} + (1 - b)\, G_{\psi_1},$$

where

$$\overline{\mathrm{Dec}_z^\kappa}(x) = \begin{cases} \mathrm{Dec}_z^\kappa(x), & x \in \alpha \bar{W}_z^\kappa, \\ 0, & \text{otherwise.} \end{cases}$$

By construction, there is a natural isometric embedding

$$M_z^\kappa(\alpha) \hookrightarrow \overline{M_z^\kappa}(\alpha).$$

This compactification allows us to invoke convergence results for Brownian motion on compact manifolds, while ensuring that in the region of interest the geometry coincides with that of the original $\kappa$-stable manifold.

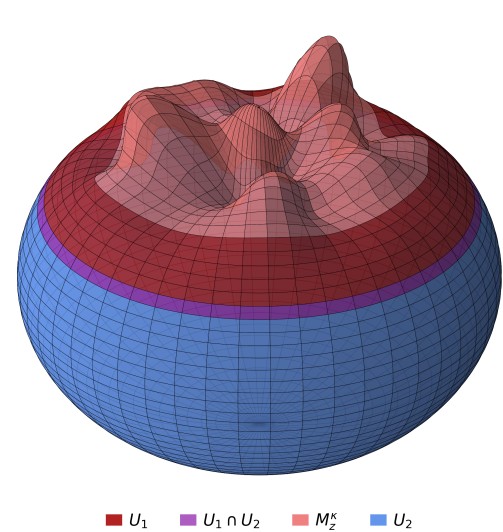

Figure 6: **Sphere-cap manifold compactification** $\overline{M_z^\kappa}$. The figure illustrates how the $M_z^\kappa$ manifold is compactified by capping off the spherical domain with the appropriate boundary conditions.

**Theorem** (Main Theorem). *Let $(Z_i^\epsilon)_{i \geq 0}$ be the sequence produced by Algorithm 1, for $M_z^\kappa(\alpha)$ with $\alpha \in (0,1)$ and diffusion horizon $T > 0$, and define its continuous-time interpolation*

$$Z^\epsilon(t) := Z^\epsilon_{\lfloor \epsilon^{-2} t \rfloor}, \qquad t \geq 0.$$

*Let $R_\kappa^z = d_{M_z^\kappa}(z, \alpha W_z^\kappa)^c)$, and suppose $L \geq 1$ satisfies*

$$\sup_{x \in M_z^\kappa(\alpha)} \mathrm{Ric}_{M_z^\kappa}(x) \geq -L^2.$$

*Then for $T < \frac{(R_z^\kappa)^2}{4 k_z^\kappa L}$, as $\epsilon \to 0$, the process $X^\epsilon$ converges in distribution to Riemannian Brownian motion stopped at the boundary of $M_z^\kappa(\alpha)$, with respect to the Skorokhod topology, on a set $C_{\kappa,z}^T \subset \Omega$ such that*

$$\mathbb{P}\big(C_{\kappa,z}^T\big) \geq 1 - \exp\Big(-\frac{(R_z^\kappa)^2}{32 T}\Big).$$

*Proof.* Let $\overline{M_z^\kappa(\alpha)}$ be the spherical-cap compactification of $M_z^\kappa(\alpha)$. Schwarz et al. (2022) showed that the non-stopped version of Algorithm 1, extended to $\overline{M_z^\kappa(\alpha)}$, produces a process $\overline{Z^\epsilon}$ that converges to $B$ in the Skorokhod topology on $\overline{M_z^\kappa(\alpha)}$.

For a closed set $A$ define the exit time

$$T_A^X = \inf\{t \in [0,T] : X(t) \notin A\} \wedge T$$

Both $T_{\alpha W_z^\kappa}^{\overline{X^\epsilon}}$ and $T_{\alpha W_z^\kappa}^B$ are valid stopping times (by right-continuity and because $(W_z^\kappa)^c$ is closed), and (from the construction of the Algorithm 1:

$$Z^\epsilon = \overline{Z^\epsilon}_{\cdot \wedge \alpha W_z^\kappa}^{\overline{X^\epsilon}}$$

Convergence in Skorokhod topology does not automatically imply convergence of stopping times (see Appendix D.1 for a counterexample). However, for the high-probability event

$$C_{\kappa,z}^T = \{\omega \in \Omega : \forall t \in [0,T], \ B_t(\omega) \in \alpha W_z^\kappa\},$$

we have that $\exists \epsilon_0$ such that $\forall \epsilon > \epsilon_0$,

$$T_{\alpha W_z^\kappa}^{\overline{X^\epsilon}} \equiv T, \quad T_{\alpha W_z^\kappa}^B \equiv T,$$

and thus convergence holds on $C_{\kappa,z}^T$.

It remains to lower-bound $\mathbb{P}(C_{\kappa,z}^T)$. Observe that

$$\Omega \setminus C_{\kappa,z}^T \subset D_{\kappa,z}^T = \left\{ \omega \in \Omega : \exists t \in [0,T], \; d_{M_z^\kappa}(B_t(\omega), z) \geq R_z^\kappa \right\}.$$

Hence

$$\mathbb{P}(C_{\kappa,z}^T) \geq 1 - \mathbb{P}(D_{\kappa,z}^T).$$

**Lemma 1** (Exit-time bound). *Suppose $L \geq 1$ satisfies*

$$\sup_{x \in M_z^\kappa(\alpha)} \mathrm{Ric}_{M_z^\kappa}(x) \geq -L^2.$$

*Let $T_{R_z^\kappa}$ be the first exit time of Riemannian Brownian motion from*

$$B_{M_z^\kappa}(B_0, R_z^\kappa) = \{ x \in M_z^\kappa(\alpha) : d_{M_z^\kappa}(B_0, x) < R_z^\kappa \}.$$

*Then*

$$\mathbb{P}(T_{R_z^\kappa} \leq T) \leq \exp\left( -\frac{(R_z^\kappa)^2}{8T}\left(1 - \frac{2Tk_z^\kappa L}{(R_z^\kappa)^2}\right)^2 \right).$$

*Proof.* Let $r(x) = d_{M_z^\kappa}(B_0, x)$ and write $r_t := r(B_t)$. On $M_z^\kappa$ the function $r$ is smooth. The semimartingale decomposition of $r_t$ (see, e.g., (Hsu, 2002a, Eq. (3.6.1))) gives

$$r_t^2 \leq 2\int_0^t r_s \, d\beta_s + \int_0^t r_s \, \Delta r(B_s) \, ds + t,$$

where $\beta$ is a real Brownian motion adapted to $B$, and we have used that the local time term is nonnegative and can be dropped to obtain an inequality.

By the Laplacian comparison theorem ($\mathrm{Ricci}(\cdot) \geq -(k_z^\kappa - 1)L^2$), for $r > 0$,

$$\Delta r \leq (k_z^\kappa - 1) L \coth(Lr) \leq (k_z^\kappa - 1)\left(L + \frac{1}{r}\right),$$

hence

$$r \, \Delta r \leq (k_z^\kappa - 1)(Lr + 1).$$

Up to the first exit time $T_{R_z^\kappa} := \inf\{t \geq 0 : r_t \geq R_z^\kappa\}$ we have $r_s \leq R_z^\kappa$, so

$$r_s \, \Delta r(B_s) \leq (k_z^\kappa - 1)(LR_z^\kappa + 1) \leq k_z^\kappa L + k_z^\kappa \leq 2 k_z^\kappa L,$$

using $L \geq 1$. Therefore, for $t = T_{R_z^\kappa} \wedge T$,

$$r_t^2 \leq 2\int_0^t r_s \, d\beta_s + 2 k_z^\kappa L t + t \leq 2\int_0^t r_s \, d\beta_s + 2 k_z^\kappa L t + t. \tag{14}$$

On the event $\{T_{R_z^\kappa} \leq T\}$ we have $t = T_{R_z^\kappa}$ and $r_t \geq R_z^\kappa$, hence from equation 14

$$(R_z^\kappa)^2 \leq 2\int_0^{T_{R_z^\kappa}} r_s \, d\beta_s + 2 k_z^\kappa L T.$$

Rearranging,

$$\int_0^{T_{R_z^\kappa}} r_s \, d\beta_s \geq \frac{(R_z^\kappa)^2 - 2 k_z^\kappa L T}{2}.$$

Set $M_t := \int_0^t r_s \, d\beta_s$, a continuous martingale with quadratic variation $\langle M \rangle_t = \int_0^t r_s^2 \, ds \leq (R_z^\kappa)^2 t$ up to time $T_{R_z^\kappa}$. By the Dambis–Dubins–Schwarz theorem there exists a standard Brownian motion $W$ such that $M_{T_{R_z^\kappa}} = W_\eta$ with $\eta = \langle M \rangle_{T_{R_z^\kappa}} \leq (R_z^\kappa)^2 T_{R_z^\kappa} \leq (R_z^\kappa)^2 T$ on $\{T_{R_z^\kappa} \leq T\}$. Thus,

$$\{T_{R_z^\kappa} \leq T\} \subset \left\{ W_\eta \geq \frac{(R_z^\kappa)^2 - 2 k\kappa(z) L T}{2} \right\}.$$

Using the Gaussian tail bound together with $\eta \leq (R_z^\kappa)^2 T$ (and the reflection principle),

$$\mathbb{P}(T_{R_z^\kappa} \leq T) \leq \exp\left( -\frac{\left((R_z^\kappa)^2 - 2 k_z^\kappa L T\right)^2}{8 (R_z^\kappa)^2 T} \right) = \exp\left( -\frac{(R_z^\kappa)^2}{8T}\left(1 - \frac{2 k_z^\kappa L T}{(R_z^\kappa)^2}\right)^2 \right),$$

which is the claimed bound. $\qquad\square$

Applying the lemma, if $T < \frac{(R_z^\kappa)^2}{4k_z^\kappa L}$, then

$$\mathbb{P}(D_{\kappa,z}^T) = \mathbb{P}(T_{R_z^\kappa} \leq T) \ \leq \ \exp\!\left(-\frac{(R_z^\kappa)^2}{32T}\right),$$

which yields

$$\mathbb{P}(C_{\kappa,z}^T) \ \geq \ 1 - \exp\!\left(-\frac{(R_z^\kappa)^2}{32T}\right).$$

$\square$

**Remark.** The lemma shows that the probability of exiting the ball of radius $R_z^\kappa$ before time $T$ decays exponentially in $\frac{(R_z^\kappa)^2}{T}$, up to curvature- and rank-dependent constants. Intuitively, this means that with overwhelming probability the Riemannian Brownian motion (and hence our random walk in the $\epsilon \to 0$ limit) remains confined inside $B_{M_z^\kappa}(z, R_z^\kappa)$ for all $t \leq T$. This high-probability control is what allows us to restrict attention to the event $C_{\kappa,z}^T$ in the proof of Theorem 1.

### D.1 ON STOPPING TIMES AND SKOROHOD CONVERGENCE

An important subtlety in the proof of Theorem 1 is that convergence of processes in the Skorohod topology does not, in general, imply convergence of associated stopping times. Figure 7 illustrates this phenomenon with a simple deterministic example.

Let $X_t = \sin(t)$, and consider the approximating sequence of processes

$$X_t^n = \left(1 - \frac{1}{n}\right) \sin(t).$$

Clearly $X^n \to X$ uniformly on compact time intervals, hence also in the Skorohod topology. However, the stopping time defined as

$$T = \inf\{t \geq 0 : X_t = 1\}$$

does not converge along this sequence. Indeed, $T = \pi/2$ for $X$, but for every finite $n$, the process $X^n$ never reaches 1 and therefore $T^n = \infty$. Thus, despite $X^n \to X$ in Skorohod topology, we have $T^n \not\to T$.

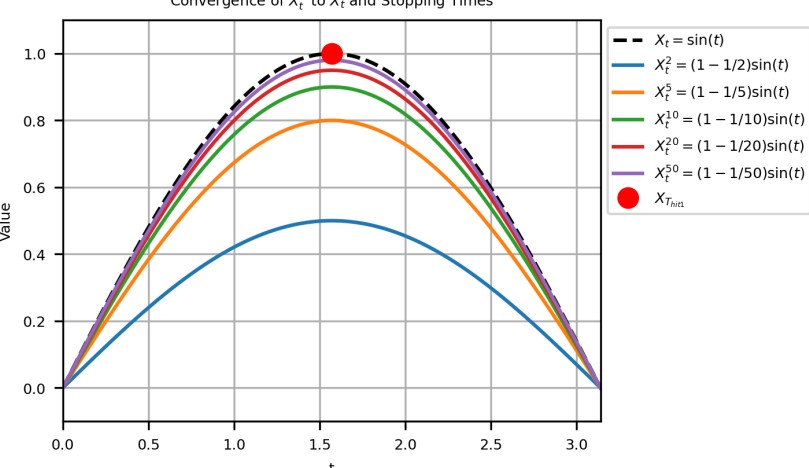

Figure 7: **Skorohod convergence does not imply convergence of stopping times.** The black dashed curve shows $X_t = \sin(t)$, which reaches 1 at $t = \pi/2$ (red dot). The colored curves show approximations $X_t^n = \left(1 - \frac{1}{n}\right) \sin(t)$, converging uniformly to $X_t$. However, none of the $X_t^n$ ever reach 1, so their stopping times for hitting level 1 are infinite. This illustrates that Skorohod convergence of processes does not guarantee convergence of stopping times defined by hitting closed sets.

In the proof of Theorem 1 we avoid this issue by restricting to the high-probability set $C_{\kappa,z}^T$ where the Brownian path remains in the interior of the ball. On this event the stopping times agree with $T$, ensuring consistency with the limiting process.

## E  SORBES IMPLEMENTATION DETAILS

We now provide practical details for the implementation of the SORBES algorithm.

**Approximating Jacobian.**  The decoder Jacobian $J_{\hat{\text{Dec}}}$ can be obtained exactly using one decoder forward pass and $LA$ backward passes, where $LA$ is a dimensionality of ambient space. To reduce computational cost, we approximate it via finite differences. Specifically, the $i$-th column of the Jacobian is approximated as

$$\frac{\partial \hat{\text{Dec}}}{\partial z_i}(z) \approx \frac{\hat{\text{Dec}}(z + \varepsilon e_i) - \hat{\text{Dec}}(z)}{\varepsilon}, \tag{15}$$

where $e_i$ denotes the $i$-th standard basis vector in the latent space $\mathcal{Z}$ and $\varepsilon > 0$ is a small perturbation parameter. This approximation requires only $d + 1$ decoder forward passes, where $d$ is a dimensionality of latent space, providing a substantial reduction in computational overhead.

**Sampling a unit tangent direction.**  We adopt the efficient implementation of Schwarz et al. (2022), which exploits a thin SVD of the decoder Jacobian to orthogonalize tangent directions.

**Approximating $\Gamma(z)[v, v]$.**  Computing Christoffel symbols directly requires evaluating first derivatives of the metric, which is computationally expensive and numerically unstable. Instead, we use an extrinsic approach: the covariant derivative of a curve can be obtained from its Euclidean acceleration in the ambient space, projected back onto the tangent space (do Carmo, 1992). Concretely, for a point $z \in \mathcal{Z}$, $z' \in W_z^\kappa$ and a probe radius $\rho > 0$, we approximate the extrinsic acceleration of a decoded curve along $v \in T_{z'} M_z^\kappa$ by a second-order central difference:

$$a_{\text{ex}}^{z,\kappa}(v; z', \rho) \approx \frac{\hat{\text{Dec}}(z' + \rho v) - 2\,\hat{\text{Dec}}(z') + \hat{\text{Dec}}(z' - \rho v)}{\rho^2}. \tag{16}$$

Projecting back to the latent tangent space using the Moore–Penrose pseudoinverse of the truncated Jacobian yields an efficient approximation of the Christoffel correction:

$$\Gamma(z')[v, v] \approx c(v; z', \rho) = J_{\hat{\text{Dec}}}^\kappa(z')^+ a_{\text{ex}}(v; z', \rho), \qquad c(v; z', \rho) \in \mathbb{R}^{k_z^\kappa}. \tag{17}$$

**Adaptive step size $\epsilon$.**  Since the computation of Christoffel symbols involves the (pseudo)inverse of the Jacobian, small values of $\kappa$ may amplify numerical noise. In this case, the update

$$z_i^\epsilon = z_{i-1}^\epsilon + \epsilon v - \epsilon^2 \Gamma[v, v] = z_{i-1}^\epsilon + \Delta(\epsilon),$$

may become unreasonably large in the ambient Euclidean metric on $U_z^\kappa$, making the algorithm unstable.

To control this, we adapt the step size $\epsilon$ so that the update norm never exceeds a predefined threshold $\Delta_{\max}$:

$$\epsilon_{\Delta_{\max}} = \min\Big\{ \epsilon' > 0 : \|\Delta(\epsilon')\| \geq \Delta_{\max} \Big\} \vee \epsilon,$$

where $a \vee b = \max\{a, b\}$. This guarantees that the step size is never greater than the nominal $\epsilon$, but shrinks adaptively whenever the second-order correction is large.

**SORBES (Stable/Efficient).**  We refer to the resulting algorithm with adaptive step size control and extrinsic approximation of Christoffel symbols (Sec. 2.3) as SORBES-SE. This variant is numerically stable in ill-conditioned regions of the decoder geometry while preserving the efficiency of the original scheme.

## F  MUTANG - MUTATION ENUMERATION IN TANGENT SPACE

The detailed description of MUTANG algorithm is presented in Algorithm 5.

---

**Algorithm 4** SORBES-SE

---

**Require:** $z \in \mathcal{Z}$, $\kappa \geq 0$, step size $\epsilon$, diffusion time $T$, maximum number of steps $\texttt{STEP}_{\max}$, $\alpha = 0.99$, stability threshold $\Delta_{\max} = 0.5$

1: $W_z^\kappa, G_{\text{Dec}} \leftarrow M_z^\kappa$
2: $z_0^\epsilon \leftarrow z$,
3: $\texttt{stopped} \leftarrow \texttt{False}$,
4: $\sigma \leftarrow 0$,          ($\sigma$ tracks diffusion time)
5: $\texttt{step} = 0$,
6: **while** $\sigma < T$ and $\texttt{step} < \texttt{STEP}_{\max}$ **do**
7:      Sample a unit tangent direction $\overline{v} \in S_z^\kappa = \{u \in T_z M_z^\kappa : \langle u, u \rangle_z^{\text{Dec}} = 1\}$
8:      Set $v \leftarrow \sqrt{k_z^\kappa}\, \overline{v}$
9:      **if** not $\texttt{stopped}$ **then**
10:        Compute trial update $\Delta(\epsilon) \leftarrow \epsilon v - \epsilon^2 \Gamma(z)[v, v]$
11:        **Adaptive adjustment:** If $\|\Delta(\epsilon)\| > \Delta_{\max}$, shrink step size:

$$\epsilon \leftarrow \min\{\epsilon' > 0 : \|\Delta(\epsilon')\| \leq \Delta_{\max}\}$$

       and recompute $\Delta(\epsilon)$.
12:        Update latent coordinate:

$$z_i^\epsilon \leftarrow z_{i-1}^\epsilon + \Delta(\epsilon)$$

13:        $\sigma \leftarrow \sigma + \epsilon^2$          (update of diffusion time)
14:        **if** $z_i^\epsilon \notin W_z^\kappa(\alpha)$ **then**
15:          $\texttt{stopped} \leftarrow \texttt{True}$
16:        **end if**
17:      **else**
18:        $z_i^\epsilon \leftarrow z_{i-1}^\epsilon$          (absorbing state)
19:      **end if**
20:      $\texttt{step} \leftarrow \texttt{step} + 1$          (step update)
21: **end while**
22: **return** $(z_i^\epsilon)_{0 \leq i \leq \texttt{step}}, \sigma$

---

**Algorithm 5** MUTANG

---

**Require:** latent $z \in \mathcal{Z}$; $\kappa \geq 0$; token threshold $\theta_{\text{tok}}$.

1: Set $U_z^\kappa, k_z^\kappa$ as in Equation 8
2: $\texttt{p} \leftarrow \texttt{p}(z)$
3: $\mathcal{P} \leftarrow \emptyset$
4: **for** $j = 1$ to $k_z^\kappa$ **do**
5:      $\Delta \text{Dec}^{(j)} \leftarrow \texttt{reshape}(U^\kappa(z)_{:,j}, (L, A))$
6:      **for** $\ell = 1$ to $L$ **do**
7:        **for** each $a \in \mathcal{A}$ **do**
8:          **if** $\left|\Delta \text{Dec}_{\ell,a}^{(j)}\right| \geq \theta_{\text{tok}}$ **then**
9:            $\mathcal{P} \leftarrow \mathcal{P} \cup \{(\ell, a)\}$
10:          **end if**
11:        **end for**
12:      **end for**
13: **end for**
14: **for** $\ell = 1$ to $L$ **do**
15:      $S_\ell \leftarrow \{a : (\ell, a) \in \mathcal{P}\} \cup \{\texttt{p}_\ell\}$          (identity included)
16: **end for**
17: $\mathcal{C}(\texttt{p}(z)) \leftarrow \prod_{\ell=1}^L S_\ell$
18: **return** $\mathcal{C}(\texttt{p}(z))$

---

## G   POGS

Below we present the details of PoGS training and evaluation metrics:

PoGS hyperparameters:

- PoGS without potential: $\lambda = 0$ and $\mu = 0.1$,
- Full PoGS: $\lambda = 0.01$ and $\mu = 0.1$.
- All: $\theta_{\text{pot}} = 5$.

PoGS metrics:

- chord ambient length:

$$\sum_{k=0}^{N-1} \|X_{k+1} - X_k\|_2$$

- chord latent length:

$$\sum_{k=0}^{N-1} \|z_{k+1} - z_k\|_2$$

For computation of seeds and wells, we excluded first and last 20% of a peptide path were excluded to avoid trivial rediscovery.

For each pair, the chord length $N$ was determined dynamically as

$$N = \lfloor \rho \cdot \|z_a - z_b\|_2 \rfloor,$$

where $\rho$ is the point density hyperparameter (set to $\rho = 90$ in our experiments). This construction guarantees that longer trajectories in the latent space are sampled more densely than shorter ones, preserving a uniform resolution across geodesics of varying length. The geodesic points $\{z_i\}_{i=1}^n$ were optimized using the Adam optimizer with learning rate $\eta = 10^{-3}$ and weight decay $10^{-5}$. We applied a `ReduceLROnPlateau` scheduler, which decreased the learning rate by a factor of $0.8$ whenever no improvement in the loss was observed for a number of iterations equal to the patience hyperparameter. The endpoints $z_a$ and $z_b$ were kept fixed throughout the optimization by zeroing their gradients at every step.

---

**Algorithm 6** POGS

---

**Require:** seeds $z_a, z_b$, potential function $\Phi$, nb of segments $N$, weights $\lambda, \mu$, steps $T$,
1: Initialize $z_0 \leftarrow z_a$, $z_N \leftarrow z_b$, $z_{1:N-1}$ by linear interpolation in latent space
2: **for** $t = 1$ to $T$ **do**
3:     $X_k \leftarrow \log(\text{Dec})(z_k)$ for $k = 0..N$
4:     Compute energy $\mathcal{E}_\lambda(Z)$ as in equation 6
5:     Take a gradient step on $z_{1:N-1}$ to minimize $\mathcal{E}_\lambda(Z)$
6: **end for**
7: **return** $\{\text{p}(z_k)\}_{k=0}^N$

---

# H  APEX-POTENTIAL FOR POGS

The APEX predictor (Wan et al., 2024) estimates minimum inhibitory concentration (MIC) values against 11 bacterial strains, but it operates on concrete peptide *sequences*. In Potential-Minimizing Geodesic Search (PoGS), optimization proceeds over *latent-space chords*, i.e., intermediate points $z$ that decode to *position-factorized distributions* over peptides rather than single sequences:

$$\text{Dec}(z) \in \mathbb{R}^{L \times A},$$

where $L$ is the maximum peptide length and $A = 21$ is the amino-acid alphabet augmented with padding. To enable PoGS, we first *distill* the sequence-level APEX potential into a surrogate that accepts peptide *distributions*.

**Dataset construction.** Peptides from the HydrAMP training set (Szymczak et al., 2023) were encoded into latent codes $z$. In order to obtain multiple distributions from a single peptide, we then created four clones $z'$ of latent codes $z$. We applied a 2×2 perturbation scheme: two clones were injected with Gaussian noise $N(0, 0.05)$, and two were left unchanged. Finally, these four latent codes were decoded to $\mathrm{Dec}(z')$, using a softmax scaling with temperature of 1.0 for one pair (noisy and non-noisy) and a temperature of 1.5 to the other pair, resulting in four distributions per peptide. This yielded 1,060,000 peptide distributions in total. For each $\mathrm{Dec}(z')$, we enumerated the $N = 20$ most-probable sequences

$$\big(P_0(z'), \ldots, P_{N-1}(z')\big) \quad \text{with probabilities} \quad \big(p_0(z'), \ldots, p_{N-1}(z')\big),$$

applied APEX to each $P_i(z')$ to obtain MIC *vectors* $\mathrm{MIC}_{P_i(z')} \in \mathbb{R}^{11}$, and defined the distribution's *expected* MIC vector via the probability-weighted average

$$\Phi_{\mathrm{MIC}}^{\mathrm{true}}\big(\mathrm{Dec}(z')\big) = \sum_{i=0}^{N-1} \mathrm{MIC}_{P_i(z')} \cdot \frac{p_i(z')}{\sum_{j=0}^{N-1} p_j(z')} \ \in \ \mathbb{R}^{11}.$$

**Training protocol and standardization.** We split the dataset into 80% train, 10% validation, and 10% test in such a way that no two sets contain distributions originating from the same peptide. Let $\mu, \sigma \in \mathbb{R}^{11}$ be the per-strain mean and standard deviation computed *on the training set*. Targets were z-scored componentwise:

$$y^z = \frac{y - \mu}{\sigma}.$$

We trained an encoder-only transformer that *operates on distributions* $\mathrm{Dec}(z) \in \mathbb{R}^{L \times A}$ and predicts z-scored MIC vectors in $\mathbb{R}^{11}$:

$$\Phi_{\mathrm{MIC}}^{\mathrm{model}} : \ \mathrm{Dec}(z) \ \mapsto \ \mathbb{R}^{11}.$$

Architecture: three transformer encoder layers (four heads), embedding dimension 128, feed-forward dimension 256, dropout 0.05. Optimization used Adam (learning rate $10^{-4}$) for 15 epochs with mean-squared error (MSE) loss on z-scored targets:

$$\mathcal{L}_{\mathrm{MSE}} = \frac{1}{11} \left\| \Phi_{\mathrm{MIC}}^{\mathrm{model}}(\mathrm{Dec}(z)) - y^z \right\|_2^2.$$

**Final potential used by PoGS.** PoGS operates on flattened *log-probabilities*. Let $X \in \mathbb{R}^{L \cdot A}$ be the flattened log-probability vector. We reconstruct a valid distribution using PyTorch-style operations:

$$P(X) = \mathrm{softmax}\big( X.\,\mathrm{reshape}(L, A), \ \mathtt{dim} = 1 \big) \ \in \ [0,1]^{L \times A},$$

where $\mathtt{dim=1}$ is the amino-acid dimension. The surrogate outputs a z-scored MIC vector

$$\widehat{m}^z(X) = \Phi_{\mathrm{MIC}}^{\mathrm{model}}\big(P(X)\big) \in \mathbb{R}^{11}.$$

Restricting to the three target *E. coli* strains (index set $\mathcal{I}_{E.\ coli}$), the scalar property potential used by PoGS is

$$\Phi(X) = \mathbf{1}^\top \big[\widehat{m}^z(X)\big]_{\mathcal{I}_{E.\ coli}}.$$

# I  LE-BO HYPERPARAMETERS

We enumerate all hyperparameter values of our optimization algorithm LE-BO and all its sub-algorithms.

- Algorithm 3 LE-BO - Local Enumeration Bayesian Optimization
  - Trust region distance $d_{\mathrm{trust}} = 2$.
  - Number of ROBOT evaluations per iteration $k_{\mathrm{ROBOT}} = 3$.
  - Diversity threshold $d_{\mathrm{ROBOT}} = 2$.
  - Following the approach of Eberhardt et al. (2024), as a surrogate model, we use a Gaussian Process GP with the Tanimoto similarity kernel Szedmak & Bach (2025), applied to the MAP4 fingerprints of peptides Capecchi et al. (2020).
  - Aquisition function GP. $\mathtt{acquistion}$ was chosen to be Log Expected Improvement.

- Algorithm 2 LOCALENUMERATION

    - $\kappa_{SORBES} = 0.01$.
    - $\kappa_{\text{MUTANG}} = 10^{-6}$.
    - Number of trajectories $M = 10$.
    - Walk time budget $T_{\text{walk}} = 0.1$.
    - Nominal step size $\epsilon = 0.1$.
    - Mutation threshold $\theta_{\text{mut}}$

- A probe radius $\rho > 0$ in the second-order central difference approximation of the extrinsic acceleration (Equation 16) $\rho = 0.05$.

- Step of the finite-difference approximation of the decoder Jacobian (Equation 15) $\varepsilon = 0.05$.

## J  WET-LAB VALIDATION

### J.1  PEPTIDE SYNTHESIS AND CHARACTERIZATION

Peptides were synthesized on an automated peptide synthesizer (Symphony X, Gyros Protein Technologies) by standard Fmoc-based solid-phase peptide synthesis (SPPS) on Fmoc-protected amino acid–Wang resins (100–200 mesh). The following preloaded resins were employed with their respective loading capacities (100 $\mu$mol scale): Fmoc-Asn(Trt)-Wang Resin (0.510 mmol g$^{-1}$), Fmoc-His(Trt)-Wang Resin (0.480 mmol g$^{-1}$), Fmoc-Leu-Wang Resin (0.538 mmol g$^{-1}$), Fmoc-Lys(Boc)-Wang Resin (0.564 mmol g$^{-1}$), Fmoc-Phe-Wang Resin (0.643 mmol g$^{-1}$), Fmoc-Thr(tBu)-Wang Resin (0.697 mmol g$^{-1}$), Fmoc-Trp(Boc)-Wang Resin (0.460 mmol g$^{-1}$), Fmoc-Tyr(tBu)-Wang Resin (0.520 mmol g$^{-1}$). In addition to preloaded resins, standard Fmoc-protected amino acids were employed for chain elongation, including: Fmoc-Ala-OH, Fmoc-Cys(Trt)-OH, Fmoc-Glu(OtBu)-OH, Fmoc-Phe-OH, Fmoc-Gly-OH, Fmoc-His(Trt)-OH, Fmoc-Ile-OH, Fmoc-Lys(Boc)-OH, Fmoc-Leu-OH, Fmoc-Met-OH, Fmoc-Asn(Trt)-OH, Fmoc-Arg(Pbf)-OH, Fmoc-Ser(tBu)-OH, Fmoc-Thr(tBu)-OH, Fmoc-Val-OH, Fmoc-Trp(Boc)-OH, and Fmoc-Tyr(tBu)-OH. N,N-Dimethylformamide (DMF) was used as the primary solvent throughout synthesis. Stock solutions included: 500 mmol L$^{-1}$ Fmoc-protected amino acids in DMF, a coupling mixture of HBTU (450 mmol L$^{-1}$) and N-methylmorpholine (NMM, 900 mmol L$^{-1}$) in DMF, and 20% (v/v) piperidine in DMF for Fmoc deprotection. After synthesis, peptides were deprotected and cleaved from the resin using a cleavage cocktail of trifluoroacetic acid (TFA)/triisopropylsilane (TIS)/dithiothreitol (DTT)/water (92.8% v/v, 1.1% v/v, 0.9% w/w, 4.8% w/w) for 2.5 hours with stirring at room temperature. The resin was removed by vacuum filtration, and the peptide-containing solution was collected. Crude peptides were precipitated with cold diethyl ether and incubated for 20 min at $-20$ °C, pelleted by centrifugation, and washed once more with cold diethyl ether. The resulting pellets were dissolved in 0.1% (v/v) aqueous formic acid and incubated overnight at $-20$ °C, followed by lyophilization to obtain dried peptides. For characterization, peptides were dried, reconstituted in 0.1% formic acid, and quantified spectrophotometrically. Peptide separations were performed on a Waters XBridge C$_{18}$ column (4.6 $\times$ 50 mm, 3.5 $\mu$m, 120 Å) at room temperature using a conventional high-performance liquid chromatography (HPLC) system. Mobile phases were water with 0.1% formic acid (solvent A) and acetonitrile with 0.1% formic acid (solvent B). A linear gradient of 1–95% B over 7 min was applied at 1.5 mL min$^{-1}$. UV detection was monitored at 220 nm. Eluates were analyzed on Waters SQ Detector 2 with electrospray ionization in positive mode. Full scan spectra were collected over m/z 100–2,000. Selected Ion Recording (SIR) was used for targeted peptides. Source conditions were capillary voltage 3.0 kV, cone voltage 25–40 V, source temperature 120 °C, and desolvation temperature 350 °C. Mass spectra were processed with MassLynx software. Observed peptide masses were compared with theoretical values, and quantitative analysis was based on integrated SIR peak areas.

### J.2  BACTERIAL STRAINS AND GROWTH CONDITIONS

The bacterial panel utilized in this study consisted of the following pathogenic strains: *Acinetobacter baumannii* ATCC 19606; *A. baumannii* ATCC BAA-1605 (resistant to ceftazidime, gentamicin, ticarcillin, piperacillin, aztreonam, cefepime, ciprofloxacin, imipenem, and meropenem);

*Escherichia coli* ATCC 11775; *E. coli* AIC221 [MG1655 phnE_2::FRT, polymyxin-sensitive control]; *E. coli* AIC222 [MG1655 pmrA53 phnE_2::FRT, polymyxin-resistant]; *E. coli* ATCC BAA-3170 (resistant to colistin and polymyxin B); *Enterobacter cloacae* ATCC 13047; *Klebsiella pneumoniae* ATCC 13883; *K. pneumoniae* ATCC BAA-2342 (resistant to ertapenem and imipenem); *Pseudomonas aeruginosa* PAO1; *P. aeruginosa* PA14; *P. aeruginosa* ATCC BAA-3197 (resistant to fluoroquinolones, $\beta$-lactams, and carbapenems); *Salmonella enterica* ATCC 9150; *S. enterica* subsp. *enterica* Typhimurium ATCC 700720; *Bacillus subtilis* ATCC 23857; *Staphylococcus aureus* ATCC 12600; *S. aureus* ATCC BAA-1556 (methicillin-resistant); *Enterococcus faecalis* ATCC 700802 (vancomycin-resistant); and *Enterococcus faecium* ATCC 700221 (vancomycin-resistant). *P. aeruginosa* strains were propagated on Pseudomonas Isolation Agar, whereas all other species were maintained on Luria-Bertani (LB) agar and broth. For each assay, cultures were initiated from single colonies, incubated overnight at 37 °C, and subsequently diluted 1:100 into fresh medium to obtain cells in mid-logarithmic phase.

### J.3 MINIMAL INHIBITORY CONCENTRATION (MIC) DETERMINATION

MIC values were established using the standard broth microdilution method in untreated 96-well plates. Test peptides were dissolved in sterile water and prepared as twofold serial dilutions ranging from 1 to 64 $\mu$mol L$^{-1}$. Each dilution was combined at a 1:1 ratio with LB broth containing $4 \times 10^6$ CFU mL$^{-1}$ of the target bacterial strain. Plates were incubated at 37 °C for 24 h, and the MIC was defined as the lowest peptide concentration that completely inhibited visible bacterial growth. All experiments were conducted independently in triplicate.

### J.4 DETAILED WET-LAB VALIDATION RESULTS

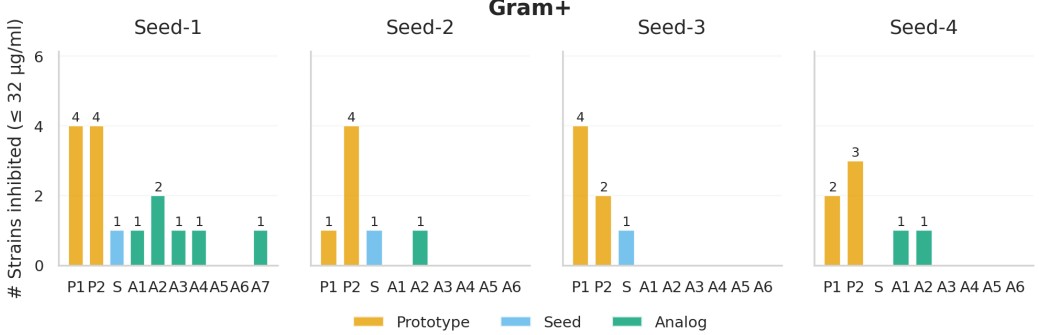

Figure 8: **Antimicrobial activity against Gram-positive bacterial strains by seed family.** Bar chart shows the number of Gram-positive strains (out of 5 total) against which each peptide achieved MIC $\leq$ 32 $\mu$g/ml, organized by seed family (Seed-1 through Seed-4). Within each family, results are shown for prototypes (P1, P2; orange), seeds (S; blue), and analogs (A1-A7; green). Numbers above bars indicate the count of active strains for each peptide.

## K REBUTTAL-RELATED ANALYSES

### K.1 SENSITIVITY ANALYSIS OF POGS

#### K.1.1 SENSITIVITY TO THE CHOICE OF METRIC

To assess sensitivity of PoGS to the choice of the metric as well as to the choice of prototypes, we compared our implementation with two alternatives: using amino-acid probabilities instead of logits, and straight (Euclidean) interpolation, on a new set of 60 prototypes (Table K.1.1). Logits yielded better performance in terms of potential, counts of seeds and wells, supporting our choice. Importantly, both metrics outperformed straight interpolation, and PoGS achieved even better results on these new prototypes than those reported in Table 1. These results show that PoGS improves results regardless of the metric used or the prototype set.

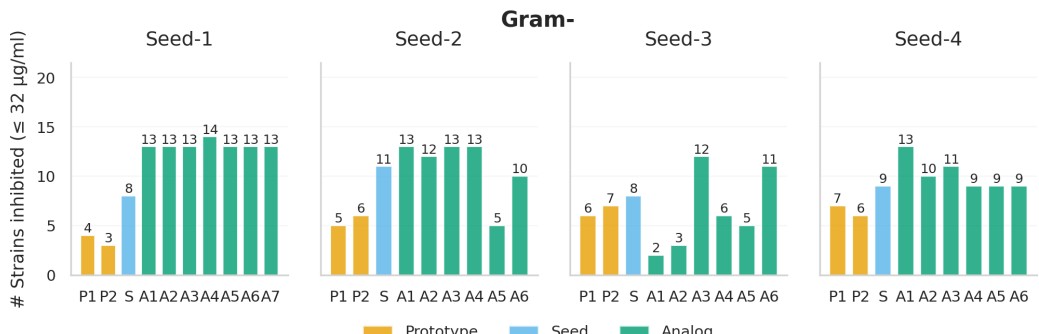

Figure 9: **Antimicrobial activity against Gram-negative bacterial strains by seed family.** Bar chart shows the number of Gram-negative strains (out of 14 total) against which each peptide achieved MIC $\leq$ 32 $\mu$g/ml, organized by seed family (Seed-1 through Seed-4). Within each family, results are shown for prototypes (P1, P2; orange), seeds (S; blue), and analogs (A1-A7; green). Numbers above bars indicate the count of active strains for each peptide.

| Method | Latent Length | Ambient Length (orig.) | Ambient Length (probs) | Peptide Path Length | Potential | Seeds | Wells |
|---|---|---|---|---|---|---|---|
| Straight interpolation | $6.50 \pm 1.34$ | $3448.7 \pm 403.8$ | $4.76 \pm 4.94$ | $47.0 \pm 20.9$ | $-1732.8 \pm 272.7$ | $117 \pm 42$ | $41 \pm 14$ |
| PoGS (with probs metric) | $11.48 \pm 3.36$ | $17227.93 \pm 147.8$ | $1.67 \pm 1.16$ | $42.7 \pm 18.1$ | $-1729.9 \pm 271.4$ | $176 \pm 43$ | $67 \pm 14$ |
| PoGS (original) | $14.12 \pm 3.07$ | $5001.4 \pm 236.7$ | $9.97 \pm 4.96$ | $63.3 \pm 29.3$ | $-2329.0 \pm 420.0$ | $188 \pm 29$ | $80 \pm 20$ |

Table 3: Table PoGS with the original metric compared to PoGS with a metric on decoded probabilities, as well as to straight interpolation, for a different set of prototypes than in Table 1.

### K.1.2 SENSITIVITY TO THE CHOICE OF POTENTIAL

To evaluate PoGS's sensitivity to the choice of potential and its applicability to multi-objective settings, we applied it to jointly maximize two physicochemical properties by assigning weight $alpha$ to the hydrophobicity and $1 - \alpha$ to charge (Table K.1.2). Across all values of $\alpha$, PoGS outperformed straight-line (Euclidean) interpolation. For this experiment, we selected a new set of 60 prototypes from the GRAMPA and DRAMP datasets, demonstrating that PoGS achieves superior performance regardless of both the potential used and the prototype set.

| Method | Latent Length | Ambient Length | Peptide Path Length | Max Hydrophobicity | Max Charge | Maximized Multiobjective Potential |
|---|---|---|---|---|---|---|
| PoGS w. $\alpha = 0.9$ | $12.07 \pm 2.57$ | $4573.57 \pm 19$ | $66.86 \pm 24.14$ | $8.40 \pm 0.72$ | $15.81 \pm 2.31$ | $24.21 \pm 2.42$ |
| PoGS w. $\alpha = 0.5$ | $10.04 \pm 2.05$ | $3624.69 \pm 185.09$ | $57.66 \pm 27.08$ | $7.64 \pm 0.63$ | $15.75 \pm 2.15$ | $23.39 \pm 2.24$ |
| PoGS w. $\alpha = 0.1$ | $10.61 \pm 2.13$ | $3651.19 \pm 190.39$ | $55.93 \pm 25.80$ | $7.91 \pm 0.64$ | $14.76 \pm 2.26$ | $22.67 \pm 2.35$ |
| Straight line (Euclidean) | $6.51 \pm 1.34$ | $3448.70 \pm 403.80$ | $47.00 \pm 20.90$ | $7.90 \pm 0.64$ | $13.76 \pm 2.22$ | $21.66 \pm 2.35$ |

Table 4: Comparison of PoGS multiobjective optimization across different $\alpha$ values and Euclidean straight-line baselines.

### K.2 LE-BO TIME AND MEMORY PROFILING

To quantify the computational gains of our method, we measured the average runtime of LE-BO over 10 iterations and 6 seeds, and compared it to SAASBO (Eriksson & Jankowiak (2021)) under the same conditions (Table K.2). LE-BO required 8× less time per iteration and used 1.5× less memory. Moreover, Local Enumeration accounted for only 35% of the total runtime, underscoring its efficiency. Average execution time of a single optimization run of LE-Bo with 1400 iterations was 1h20m (±30m). All computations were performed for the HydrAMP model with dimension 64 and measured on a Mac Mini M4Pro machine with 24GB of RAM.

| Method | Local Enumeration Time / Iteration (s) | Total Iteration Time (s) | Memory (MB) |
|---|---|---|---|
| LE-BO | $1.03 \pm 0.31$ | $2.89 \pm 2.21$ | $1490 \pm 210$ |
| SAASBO (Eriksson & Jankowiak (2021)) | N/A | $23.31 \pm 9.94$ | $2248 \pm 12$ |

Table 5: Runtime and memory comparison between LE-BO and SAASBO (Eriksson & Jankowiak (2021)).

### K.3 LE-BO ABLATION STUDY

#### K.3.1 ANALYSIS OF RESULTS FOR DIFFERENT $\kappa_{SORBES}$ AND $\kappa_{MUTANG}$

Additional ablation study show that alternative $\kappa$ values can even outperform those originally selected, demonstrating that LE-BO's performance can be further enhanced through targeted hyperparameter tuning $K.3.1$. An ablation with respect to $\alpha$ is unnecessary because, with only a single SORBES-SE step, $\alpha$ has no effect on the search process.

| $\kappa_{\text{mutang}}$ | $\kappa_{\text{sorbes}}$ | FL14 | KY14 | KF16 | KK16 | mammuthusin-3 | hydrodamin-2 | Avg. Diff. from LE-BO |
|---|---|---|---|---|---|---|---|---|
| $10^{-8}$ | $10^{-3}$ | $0.74 \pm 0.28$ | $0.83 \pm 0.44$ | $0.56 \pm 0.29$ | $0.58 \pm 0.30$ | $\mathbf{0.44 \pm 0.29}$ | $\mathbf{0.38 \pm 0.13}$ | $0.04 \pm 0.18$ |
| $10^{-4}$ | $10^{-3}$ | $0.53 \pm 0.26$ | $0.65 \pm 0.39$ | $0.66 \pm 0.11$ | $\mathbf{0.36 \pm 0.09}$ | $0.60 \pm 0.69$ | $0.45 \pm 0.14$ | $\mathbf{-0.01 \pm 0.12}$ |
| $10^{-8}$ | $10^{-1}$ | $\mathbf{0.41 \pm 0.11}$ | $0.65 \pm 0.44$ | $\mathbf{0.38 \pm 0.14}$ | $0.73 \pm 0.34$ | $0.58 \pm 0.21$ | $0.45 \pm 0.11$ | $-0.01 \pm 0.19$ |
| $10^{-4}$ | $10^{-1}$ | $0.83 \pm 0.20$ | $0.54 \pm 0.31$ | $0.72 \pm 0.29$ | $0.66 \pm 0.31$ | $0.60 \pm 0.45$ | $0.63 \pm 0.40$ | $0.12 \pm 0.07$ |
| $10^{-6}$ (orig.) | $10^{-2}$ (orig.) | $0.604 \pm 0.22$ | $\mathbf{0.502 \pm 0.24}$ | $0.600 \pm 0.29$ | $0.498 \pm 0.14$ | $0.498 \pm 0.38$ | $0.581 \pm 0.34$ | $0.0 \pm 0.0$ |

Table 6: Table LE-BO performance for different hyperparameter values. Best values of minimized MIC as predicted by APEX are bolded. The last column reports the average difference between the alternative hyperparameter setting and the original used in the manuscript.

**Analysis of results of LE-BO without ROBOT (Maus et al. (2023))**

Ablation analysis of the model without ROBOT Maus et al. (2023) confirms that ROBOT improves LE-BO for 4 out of 6 seeds (with average difference in log MIC of -0.03).

| Method | FL14 | KY14 | KF16 | KK16 | mammuthusin-3 | hydrodamin-2 | Avg. diff. from LE-BO |
|---|---|---|---|---|---|---|---|
| LE-BO w/o ROBOT | $0.75 \pm 0.41$ | $0.52 \pm 0.17$ | $0.56 \pm 0.11$ | $0.63 \pm 0.28$ | $0.57 \pm 0.61$ | $0.42 \pm 0.20$ | $0.03 \pm 0.12$ |
| LE-BO | $\mathbf{0.604 \pm 0.22}$ | $\mathbf{0.502 \pm 0.24}$ | $\mathbf{0.600 \pm 0.29}$ | $\mathbf{0.498 \pm 0.14}$ | $\mathbf{0.498 \pm 0.38}$ | $\mathbf{0.581 \pm 0.34}$ | $\mathbf{0.000 \pm 0.000}$ |

Table 7: Comparison of LE-BO with and without ROBOT across six peptides and mean deviation from LE-BO baseline.

#### K.3.2 ANALYSIS OF RESULTS FOR OTHER SEEDS

To show consistent gains compared to ablations across different seeds, we additionally performed an LE-BO ablation study on three further seeds (LL13, RC16, and KI21) from the PoGS optimization (Table K.3.2). The results confirm that clear advantage of enabling mutations and of using SORBES-SE as the random walk method can be observed regardless of the chosen seeds. Taken together, all three LE-BO components - SORBES-SE, enabling mutation (MUTANG), and ROBOT - contribute meaningful improvements. However, enabling mutation with MUTANG provides the most substantial benefit: across all ablations, the LE-BO variants incorporating MUTANG consistently outperform their counterparts without it.

| | Walk | Mutation | LL13 | RC16 | KI21 | Difference from LE-BO |
|---|---|---|---|---|---|---|
| | Euclidean | $\times$ | $0.787 \pm 0.408$ | $1.141 \pm 0.298$ | $1.170 \pm 0.283$ | $1.033 \pm 0.174$ |
| | SORBES-SE | $\times$ | $1.009 \pm 0.408$ | $1.128 \pm 0.298$ | $1.120 \pm 0.283$ | $0.709 \pm 0.204$ |
| | – | $\checkmark$ | $1.343 \pm 0.356$ | $1.225 \pm 0.275$ | $0.803 \pm 0.238$ | $0.747 \pm 0.125$ |
| | Euclidean | $\checkmark$ | $0.630 \pm 0.190$ | $0.751 \pm 0.291$ | $0.505 \pm 0.199$ | $0.253 \pm 0.091$ |
| LE-BO | SORBES-SE | $\checkmark$ | $\mathbf{0.482 \pm 0.226}$ | $\mathbf{0.458 \pm 0.243}$ | $\mathbf{0.190 \pm 0.191}$ | $\mathbf{0.000 \pm 0.000}$ |

Table 8: Confirmation of improved performance of LE-BO compared to ablations for different seeds than in Table 3.3

#### K.3.3 ANALYSIS OF RESULTS FOR DIFFERENT ORACLES

To evaluate robustness across different oracles and to verify that additional peptide properties benefit from geometry-aware exploration, we successfully applied LE-BO and show that geometry-aware exploration improves prototype peptides for three tasks:

1. minimize MIC using DEEP-AMP Pandi et al. (2023) regressor other than APEX as oracle,

2. minimize toxicity using ToxiPrep Guan et al. (2025) classification probabilities as oracle,

3. maximize hydrophobicity computed in the Eisenberg scale Eisenberg et al. (1982) as oracle.

| | KY14 | KF16 | KK16 | FL14 | mammuthusin-3 | hydrodamin-2 |
|---|---|---|---|---|---|---|
| $\log_2$(MIC) (seed) | 2.00 | 5.06 | 1.85 | 1.82 | 4.02 | 5.09 |
| Euclidean LE-BO | -0.66 ± 0.50 | 0.04 ± 0.19 | -0.56 ± 0.65 | -0.55 ± 0.46 | -0.48 ± 0.72 | -0.15 ± 0.56 |
| LE-BO | **-0.80 ± 1.06** | **-0.47 ± 0.38** | **-1.20 ± 1.23** | **-2.35 ± 0.41** | **-0.77 ± 0.75** | **-0.93 ± 0.89** |

Table 9: Minimization of $\log_2$(MIC) values using LE-BO with DEEP-AMP Pandi et al. (2023) as oracle.

| | KY14 | KF16 | KK16 | FL14 | mammuthusin-3 | hydrodamin-2 |
|---|---|---|---|---|---|---|
| Toxicity (seed) | 0.8789 | 0.9283 | 0.9044 | 0.9932 | 0.5193 | 0.0522 |
| Euclidean LE-BO | 0.014 ± 0.002 | 0.016 ± 0.003 | 0.016 ± 0.005 | 0.015 ± 0.001 | 0.014 ± 0.002 | 0.016 ± 0.002 |
| LE-BO | **0.0115 ± 0.0007** | **0.0117 ± 0.0012** | **0.0120 ± 0.0022** | **0.0122 ± 0.0015** | **0.0126 ± 0.0013** | **0.0135 ± 0.0012** |

Table 10: Table Minimization of toxicity values using LE-BO with toxicity probabilities returned by ToxiPep Pandi et al. (2023) as oracle.

| | KY14 | KF16 | KK16 | FL14 | mammuthusin-3 | hydrodamin-2 |
|---|---|---|---|---|---|---|
| Hydrophobicity (seed) | $-0.34$ | $-0.60$ | 0.03 | 0.12 | 0.17 | $-0.33$ |
| Euclidean LE-BO | 1.114 ± 0.091 | 1.208 ± 0.048 | 1.162 ± 0.112 | 1.193 ± 0.029 | 1.323 ± 0.031 | **0.963 ± 0.084** |
| LE-BO | **1.255 ± 0.013** | **1.270 ± 0.048** | **1.242 ± 0.067** | **1.265 ± 0.026** | **1.325 ± 0.022** | 0.953 ± 0.095 |

Table 11: Table Maximization of hydrophobicity using LE-BO with hydrophobicity values computed in Eisenberg scale Eisenberg et al. (1982) oracle.

### K.4 ANALYSIS OF THE $\kappa$-STABLE DIMENSION W.R.T. TO OPTIMIZATION AND PHYSICOCHEMICAL PROPERTIES

As evaluated for two models, HydrAMP and PepCVAE, the effective rank is almost always much lower than the latent dimension (see distribution of the effective rank quantified in Fig. 4A, B). Moreover, it strongly positively correlates with peptide length (see Fig. 4C,D in Appendix $B$).

Our new analysis in Figure 10 in Appendix K shows that LE-BO with SORBES-SE and mutation enabled (Fig 10 A, E) identifies more candidates in Local Enumeration for higher k-stable dimensions / longer peptides than for lower dimensions / shorter peptides. Euclidean-based search exhibits the opposite trend (Fig. 10, B,C,F,G). As our optimization method relies on dense populations of peptide neighborhoods, this explains weaker performance of Euclidean approaches. Moreover, LE-BO with SORBES-SE and mutation produces substantially more candidates on average (16 730 ± 14 786; Fig. 10 A, E) than the Euclidean search with mutation (442 ± 151; Fig. 10 B, F). Even without mutation, SORBES-SE yields higher candidate counts (17 ± 5; Fig. 10, Panels D, H) than its Euclidean counterpart (13 ± 5; Fig. 10, Panels C, G).

Moreover, the $\kappa$-stable dimension of the latent space is strongly negatively correlated with charge (Spearman r = -0.55) and aromaticity (r = -0.57) (Fig. 11). This indicates that peptides with high latent dimension tend to have higher solubility, but lower membrane affinity and antimicrobial activity. As LE-BO identifies substantially more candidates than Euclidean methods in these high-dimensional regions, increasing the likelihood that the unfavorable antimicrobial properties can be corrected.

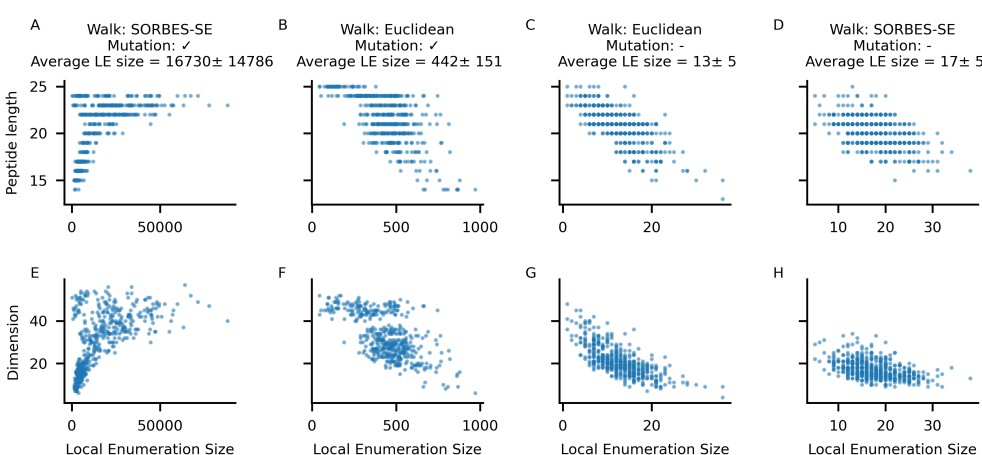

Figure 10: **Relationships between Local Enumeration (LE) result size and sequence properties under different ablations of LE-BO walk and mutation strategies**. Each panel (**A–H**) shows scatter plots comparing LE size with either peptide length (top row) or $\kappa$-stable dimension (bottom row) for four experimental LE-BO conditions: SORBES-SE walk with mutation (default LE-BO implementation) (**A, E**), Euclidean walk with mutation (**B, F**), Euclidean walk without mutation (**C, G**), and SORBES-SE walk without mutation (**D, H**). Titles report the mean ± standard deviation of the LE size for each condition. Together, these comparisons highlight how walk geometry and mutation choice influence the size and structure of local neighborhoods explored during Local Enumeration.

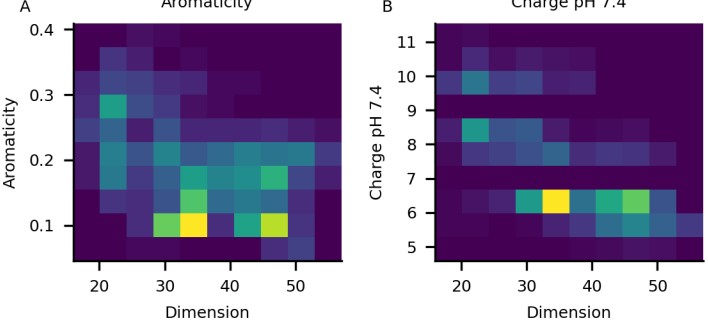

Figure 11: **Relationship between peptide physicochemical properties and latent dimension**. (**A**) Aromaticity versus embedding dimension, showing that peptides with lower aromatic residue content tend to occupy higher-dimensional regions of the embedding space. (**B**) Net charge at pH 7.4 versus embedding dimension, highlighting a similar trend in which peptides with lower cationic charge are more common at higher dimensions.

Table 12: Bacterial strains used for experimental validation of antimicrobial peptide libraries. Strains marked with MDR are multidrug-resistant clinical isolates.

| ID | Bacterial Strain |
|---|---|
| AB1 | *A. baumannii* ATCC 19606 |
| AB2$_{MDR}$ | *A. baumannii* ATCC BAA-1605 |
| EC1 | *E. cloacae* ATCC 13047 |
| EC2 | *E. coli* ATCC 11775 |
| EC3 | *E. coli* AIC221 |
| EC4$_{MDR}$ | *E. coli* AIC222 |
| EC5$_{MDR}$ | *E. coli* ATCC BAA-3170 |
| KP1 | *K. pneumoniae* ATCC 13883 |
| KP2$_{MDR}$ | *K. pneumoniae* ATCC BAA-2342 |
| PA1 | *P. aeruginosa* PAO1 |
| PA2 | *P. aeruginosa* PA14 |
| PA3$_{MDR}$ | *P. aeruginosa* ATCC BAA-3197 |
| SE1 | *S. enterica* ATCC 9150 |
| SE2 | *S. enterica Typhimurium* ATCC 700720 |
| BS1 | *B. subtilis* ATCC 23857 |
| SA1 | *S. aureus* ATCC 12600 |
| SA2$_{MDR}$ | *S. aureus* ATCC BAA-1556 |
| EFS1$_{MDR}$ | *E. faecalis* ATCC 700802 |
| EFU1$_{MDR}$ | *E. faecium* ATCC 700221 |

Table 13: Minimum inhibitory concentration (MIC, in $\mu$mol L$^{-1}$) values and peptide sequences for PepCompass-generated peptides tested against bacterial pathogen panel. '-' indicates MIC >64 $\mu$mol L$^{-1}$. Strain IDs correspond to Table 12.

| ID | Sequence | AB1 | AB2 | EC1 | EC2 | EC3 | EC4 | EC5 | KP1 | KP2 | PA1 | PA2 | PA3 | SE1 | SE2 | BS1 | SA1 | SA2 | EFS1 | EFU1 |
|---|---|---|---|---|---|---|---|---|---|---|---|---|---|---|---|---|---|---|---|---|
| Prototype-1-a | ILRWKKRKLVWKR | 64 | - | - | 16 | 64 | 32 | - | - | - | 8 | 32 | - | - | - | - | 32 | 32 | - | - |
| Prototype-1-b | FLILRWSRFARVLL | 8 | - | - | 64 | 8 | 16 | - | - | - | - | - | - | - | - | - | 16 | 8 | 32 | 4 |
| Seed-1 | FLYKWWIRIGRLKL | 1 | 1 | - | 32 | 4 | 8 | 16 | - | - | - | - | - | 32 | 8 | 16 | - | - | 64 | 64 |
| LE-BO-1-1 | RYAKINLRTAWRKLKWLIKKVMKKW | 4 | 4 | 64 | 8 | 4 | 4 | 4 | 8 | 32 | 8 | 4 | 4 | 2 | 2 | - | 32 | - | - | - |
| LE-BO-1-2 | RYAKINLRTAWRKLKWLIKKVMKWW | 8 | 8 | 64 | 8 | 4 | 16 | 8 | 8 | 8 | 8 | 8 | 8 | 4 | 4 | 64 | 16 | 32 | - | - |
| LE-BO-1-3 | RKANLKSRYAWLKLRKLIKALIAWK | 2 | 4 | - | 8 | 4 | 4 | 4 | 4 | 8 | 8 | 4 | 4 | 2 | 2 | - | 32 | 64 | - | - |
| LE-BO-1-4 | RKANLKSRYAWLKLRKLIKALVAWK | 1 | 16 | 32 | 4 | 2 | 4 | 4 | 4 | 8 | 4 | 4 | 4 | 2 | 2 | - | 32 | - | - | - |
| LE-BO-1-5 | RKANLKSRYAWLKLRKLIKAVILWK | 4 | 8 | - | 16 | 8 | 8 | 4 | 16 | 8 | 8 | 4 | 8 | 4 | 2 | - | - | - | - | - |
| LE-BO-1-6 | RKANLKTRYAWLKLRKLIKAVVNWK | 1 | 2 | - | 16 | 4 | 8 | 4 | 4 | 8 | 8 | 4 | 2 | 2 | 2 | - | 64 | - | - | - |
| LE-BO-1-7 | RKANLKIRYAWLKLRNLIKAAINWK | 1 | 1 | - | 4 | 4 | 4 | 2 | 2 | 2 | 8 | 8 | 16 | 2 | 4 | - | - | 16 | - | - |
| Prototype-2-a | KFWARGRKPWKLAIQILK | 4 | - | - | 8 | 4 | 2 | - | - | - | - | 16 | - | - | - | - | - | - | - | 4 |
| Prototype-2-b | ILRWKKRWKVWLR | 8 | - | - | 2 | 2 | 1 | - | - | - | 2 | 8 | - | - | - | - | 16 | 16 | - | - |
| Seed-2 | KFRNRHRWKFKLIFRN | 4 | 8 | - | 16 | 16 | 32 | 8 | - | - | 8 | 8 | 16 | 8 | 4 | - | - | - | - | 16 |
| LE-BO-2-1 | NRRKYLRYWLKKLLRKILKAAINAW | 8 | 8 | - | 16 | 4 | 16 | 16 | 4 | 8 | 8 | 16 | 8 | 4 | 4 | - | - | - | - | - |
| LE-BO-2-2 | KARIKLYYRWKLKLKWLLKAMIKAW | 8 | 8 | - | 16 | 4 | 16 | 8 | 8 | 4 | 8 | 16 | 64 | 8 | 32 | 32 | - | - | - | - |
| LE-BO-2-3 | KARIKLRYRWKLKLKWLLKMAAMAW | 1 | 1 | 64 | 2 | 1 | 2 | 2 | 1 | 1 | 2 | 1 | 4 | 1 | 1 | - | - | - | - | - |
| LE-BO-2-4 | KARIKLRYRWKLKLKWLLKAMMAAW | 2 | 1 | - | 2 | 2 | 2 | 4 | 4 | 8 | 4 | 2 | 4 | 4 | 4 | - | - | - | - | - |
| LE-BO-2-5 | KARIKLRYRWKLKLKWLLKMAWAAW | 32 | 64 | - | 64 | 32 | 64 | 16 | 16 | 64 | 16 | 64 | - | - | - | - | - | - | - | - |
| LE-BO-2-6 | KARIKLRYRWRLKLKWLLKAMFAW | 4 | 16 | - | 16 | 8 | 16 | 16 | 16 | - | - | 16 | 16 | 16 | 64 | - | - | 16 | 32 | - |
| Prototype-3-a | ILRWKFRKWVWLR | 4 | - | - | 2 | 8 | 2 | - | - | - | 8 | 8 | - | - | - | - | 16 | 32 | - | - |
| Prototype-3-b | WRHKSLWIRKYLKNLALLA | 0.78 | - | - | 3.12 | 0.78 | 1.56 | - | 25 | - | 6.25 | - | - | - | - | - | 50 | 25 | - | 3.12 |
| Seed-3 | KKYWLIRKWIRLWFLT | 16 | 32 | - | 16 | 32 | 8 | 64 | 64 | - | 32 | 64 | 16 | 32 | 16 | 64 | - | - | - | - |
| LE-BO-3-1 | KKARNLRKWAYLKYRLKLKILAINW | 32 | 64 | - | - | 8 | - | 64 | 64 | - | 64 | - | 64 | - | - | - | - | - | - | - |
| LE-BO-3-2 | KKARNLRWKAYLKYRLKLKILAWNK | 8 | 8 | - | 64 | 64 | - | - | - | - | - | - | 32 | 64 | - | - | - | - | - | - |
| LE-BO-3-3 | KKRRKLTLKLKLKKLLRLL | 2 | 1 | 64 | 2 | 2 | 4 | 4 | 64 | 8 | 4 | 1 | 4 | 4 | 8 | - | - | - | - | - |
| LE-BO-3-4 | KKARNLRKWAYLKYRLKLKILAANW | 32 | 32 | - | - | 4 | - | 16 | 64 | - | - | - | 16 | 32 | 64 | - | - | - | - | - |
| LE-BO-3-5 | KLRISLKARWRLWKMYVLKWKAAIW | - | - | - | 32 | 2 | 16 | 64 | 16 | 16 | 64 | - | - | 64 | - | - | - | - | - | - |
| LE-BO-3-6 | KKRRILRKWTRLWKKLLELMAAWFH | 8 | - | - | 32 | 4 | 4 | 8 | - | 8 | 32 | 8 | 8 | 8 | 16 | - | - | - | - | - |
| Prototype-4-a | LIRWKVRWLAFRRL | 4 | - | - | 4 | 8 | 8 | - | 16 | - | 16 | 16 | - | - | - | - | - | 64 | 32 | 1 |
| Prototype-4-b | KYCWRWFKLLFKKL | 4 | - | - | 4 | 2 | 2 | - | 8 | - | - | 32 | - | - | - | - | 16 | 16 | - | 4 |
| Seed-4 | KYCRRFRWLTFRWL | 64 | 32 | - | 16 | 8 | 8 | 16 | 16 | 32 | 64 | 64 | 64 | 16 | 16 | 64 | - | - | - | - |
| LE-BO-4-1 | KRARNYYRWKLWKKLKILLKAAMAW | 2 | 2 | - | 8 | 4 | 4 | 8 | 2 | 4 | 8 | 8 | 8 | 8 | 8 | - | 32 | - | - | - |
| LE-BO-4-2 | KRIRKLRILRTWKWWKLEMAAAFH | 16 | 4 | - | 64 | 8 | 16 | 32 | 32 | - | 64 | 16 | 32 | 4 | 32 | 64 | - | - | - | 32 |
| LE-BO-4-3 | KRLRKLRILRTWKWWKLEMAAAFH | 16 | 16 | - | 32 | 16 | 16 | 16 | 16 | - | 64 | 32 | 16 | 2 | 8 | - | - | - | - | 64 |
| LE-BO-4-4 | KRIRKLRILRTWKWWKLEMAMAFH | 16 | 8 | - | 64 | 16 | 4 | 8 | 16 | - | 64 | 16 | 64 | 4 | 8 | 64 | - | - | - | - |
| LE-BO-4-5 | KRLRKLRILRTWKWWKLEMAAAFHY | 32 | 16 | - | - | 8 | 16 | 16 | 64 | - | - | 16 | 16 | 4 | 16 | - | - | - | - | - |
| LE-BO-4-6 | KRLRKLRILRTWKWWKLEMAAAFHF | 4 | 4 | - | 16 | 8 | 4 | 16 | 16 | - | - | - | 16 | 4 | - | - | - | - | - | 64 |

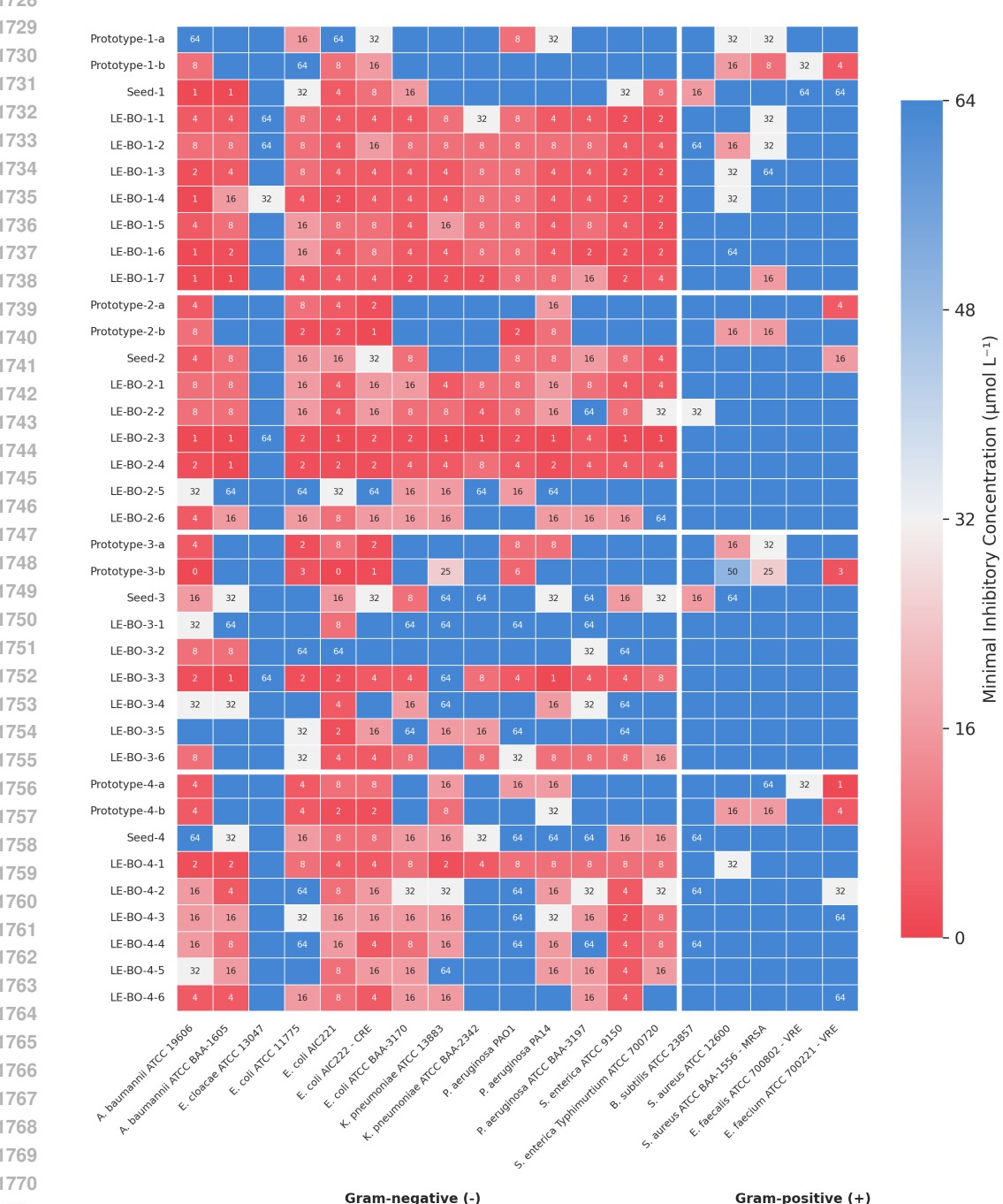

Figure 12: **Minimum inhibitory concentration profiles of antimicrobial peptide libraries against Gram-negative and Gram-positive bacterial pathogens**. MIC values (in $\mu$mol L$^{-1}$) for 37 peptide sequences evaluated against 19 bacterial strains. Peptides are stratified by seed family (seed-1 through seed-4), comprising parental prototype sequences and corresponding analogs generated via LE-BO. The bacterial panel encompasses 14 Gram-negative strains including carbapenem-resistant *Enterobacteriaceae* (CRE: *E. coli* AIC222 and ATCC BAA-3170), extended-spectrum $\beta$-lactamase-producing *K. pneumoniae* (ATCC BAA-2342), and fluoroquinolone-resistant *P. aeruginosa* (ATCC BAA-3197), alongside 5 Gram-positive strains including methicillin-resistant *S. aureus* (MRSA: ATCC BAA-1556) and vancomycin-resistant *Enterococcus* species (VRE: *E. faecalis* ATCC 700802, *E. faecium* ATCC 700221).

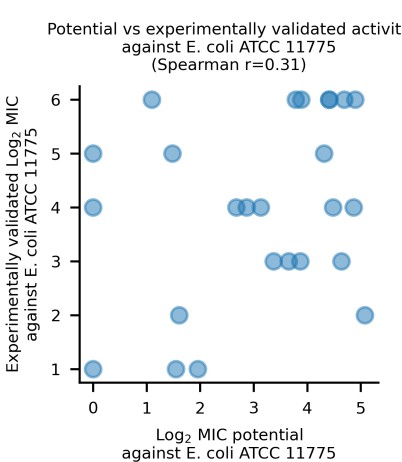

Figure 13: **Correlation between APEX potential and experimental antimicrobial activity for 29 validated peptides.** LE-BO–predicted $Log_2$ MIC potential shows a meaningful positive association with experimentally measured $Log_2$ MIC against *E. coli* ATCC 11775 (Spearman $r = 0.31$).

