# OpenReview forum: "PepCompass: Navigating Peptide Embedding Spaces Using Riemannian Geometry"
_ICLR.cc/2026/Conference — ICLR 2026 Conference Desk Rejected Submission_

### Official Review · Reviewer_i34e · 2025-10-26

**Soundness:** 3
**Presentation:** 3
**Contribution:** 2
**Rating:** 6
**Confidence:** 2

**Summary:**

This paper introduces "PepCompass", a novel framework for navigating the latent spaces of generative models for peptide design. The core contribution is a geometry-aware approach that addresses the limitations of standard Euclidean exploration. The authors first define a "Union of κ-Stable Riemannian Manifolds" to model the latent space more faithfully, accounting for varying local dimensionality. They then propose two key methods for exploration: SORBES, a second-order approximation of Riemannian Brownian motion for local sampling, and MUTANG, which interprets tangent directions as discrete amino-acid mutations. These are combined into a Local Enumeration Bayesian Optimization (LE-BO) algorithm. For global exploration, they introduce Potential-minimizing Geodesic Search (PoGS), which finds promising "seed" peptides by interpolating between prototypes along property-enriched geodesics. The method is validated with impressive wet-lab results, reporting a 100% success rate in generating active antimicrobial peptides (AMPs).

**Strengths:**

Novelty and Technical Depth: The paper makes significant contributions at the intersection of geometric deep learning and biological sequence design. The formulation of the κ-stable manifold union is a principled and elegant solution to the problem of varying intrinsic dimensionality in latent spaces. The SORBES algorithm, with its second-order correction and associated convergence theorem, represents a substantial advance over naive first-order random walks.

Comprehensive Methodology: The framework is well-rounded, offering tools for both global exploration (PoGS) and local optimization (LE-BO). The MUTANG method is particularly clever, bridging the gap between continuous latent geometry and discrete sequence space in an interpretable way.

Compelling Empirical Validation: The most striking strength is the successful in vitro validation. Achieving a 100% success rate (29/29 peptides active) and discovering 25 highly active, broad-spectrum AMPs against multidrug-resistant strains is a powerful demonstration of the method's real-world utility. This goes far beyond the typical computational benchmarks in the field.

Rigorous Ablations: The paper includes thorough ablation studies for LE-BO, clearly demonstrating the individual contributions of the Riemannian walk (SORBES) and the tangent-space mutations (MUTANG).

**Weaknesses:**

1. Reproducibility and Presentation of Wet-Lab Experiments: The presentation of the experimental results is confusing and lacks critical details, which undermines an otherwise stellar validation.

1.1 Missing Sequences: All peptide sequences are not listed in the main text or the appendices. This is a major omission for a paper centered on peptide design, as it prevents replication, analysis, and a proper understanding of the generated molecules.

1.2 Incomplete Results: Figure 9 shows only 6-7 analogs per seed family, but the text states 25 analogs were generated for each seed. It is unclear how these 25 are distributed across the seeds and why only a subset is visualized. A complete table of all analog peptides and their MIC values against all 19 strains is essential.

1.3 Justification for Bacterial Panel: The optimization objective targeted E. coli activity (a Gram-negative bacterium). However, the validation panel includes 14 Gram-negative and 5 Gram-positive strains. The rationale for this broader panel is not well-explained. The results in Figure 8 show that performance on Gram-positive bacteria is poor and, in many cases, worse than the prototypes. This suggests that the optimization successfully specialized for Gram-negative activity, but the authors should explicitly discuss this point and the implications of this specificity.

2. Computational Complexity: While the paper mentions the computational efficiency of LE-BO compared to other Bayesian optimization methods, the cost of computing the κ-stable manifolds, SVDs, and second-order corrections for SORBES is non-trivial. A more detailed discussion of the computational budget and scaling would be helpful for practitioners.

**Questions:**

1. Where are the sequences for all wet-lab validated peptides? Please provide them in a table, along with their full MIC profiles against all tested strains.

2. Please clarify the discrepancy between the 25 generated analogs and the subset shown in Figures 8 and 9. Are the shown analogs the best-performing ones? If so, state this clearly. Presenting only a curated subset can be misleading.

3. The optimization targeted E. coli, yet validation included Gram-positive strains. Could you please discuss the rationale for this choice and explicitly address the observation that Gram-positive activity did not improve (and often degraded) during optimization? Does this indicate a potential trade-off or specialization effect?

4. The PoGS potential function is a distilled model from APEX. How sensitive are the discovered seeds to the accuracy of this surrogate model? A brief analysis of the correlation between the predicted potential and the final wet-lab MIC would strengthen this section.

---

> ### Author Response · Authors · 2025-11-22
> **Answer to a review (Part 1/2)**
>
> Thank you for the thoughtful and careful review. We address each weakness (W) and question (Q) point-by-point below and refer to the updated figures in the resubmitted PDF, Appendix K.
>
> ## W1: Reproducibility and Presentation of Wet-Lab Experiments
>
> Please see answers to Q1, Q2 and Q3
>
> ## W2: Computational Complexity
>
> To quantify the computational complexity of our method, we measured the average runtime of LE-BO over 10 iterations and 6 seeds, and compared it to SAASBO under the same conditions. LE-BO required 8× less time per iteration and used 1.5× less memory.
> The cost of computing κ-stable manifolds, SVDs, and second-order corrections for SORBES is included in Local Enumeration, which accounted for only 35% of the total runtime, underscoring its efficiency. Average execution time of a single optimization run of LE-B0 with 1400 iterations was 1h20m (±30m). All computations were performed for the HydrAMP model with dimension 64 and measured on a Mac Mini M4Pro machine with 24GB of RAM.
>
> | Method | Local Enumeration Time / Iteration | Total Iteration Time (s) | Memory (MB) |
> |--------|------------------------------------|----------------------------|-------------|
> | LE-BO  | 1.03 ± 0.31                         | **2.89 ± 2.21**               | **1490 ± 210**  |
> | SAASBO | N/A                                 | 23.31 ± 9.94              | 2248 ± 12   |
>
> ## Q1: Sequences for all wet-lab validated peptides
>
> Thank you for your effort towards reproducibility. As requested, we provide all sequences of tested peptides with MIC values in Appendix K (Figure 12, Table 4 and Table 5). In the initial submission, we chose not to disclose peptide sequences following the precedent of prior AI conference works that also omit sequences of designed molecules (e.g., [1,2]).
>
> ##  Q2: Please clarify the discrepancy between the 25 generated analogs and the subset shown in Figures 8 and 9. Are the shown analogs the best-performing ones? If so, state this clearly. Presenting only a curated subset can be misleading.
>
> Thank you for raising this point. To clarify, our discovery pipeline operates at three levels:
>
> * **Prototypes (P1, P2):** Known AMPs that serve as anchors for geodesic interpolation.  These are not novel peptides and therefore not counted among the 29 experimentally evaluated peptides.
>
> * **Seeds (S):** Novel peptides discovered via PoGS through geodesic interpolation between prototype pairs. We identified 4 such seeds for experimental validation.
>
> * **Analogs ($A_1$ - $A_n$):** Novel peptides derived from each seed through LE-BO optimization to enhance activity against E. coli. We generated a total of 25 analogs across the 4 seed families.
>
> Regarding the reviewer's concern about Figure 9: the figure displays all 25 analogs distributed as follows:
> * Seed-1 → 7 analogs (LE-BO-1-1 -LE-BO-1-7)
> * Seed-2 → 6 analogs (LE-BO-2-1 -LE-BO-2-6)
> * Seed-3 → 6 analogs (LE-BO-3-1 -LE-BO-3-6)
> * Seed-4 → 6 analogs (LE-BO-4-1 -LE-BO-4-6)
>
> The phrase '25 analogs derived from these 4 seeds' refers to 25 analogs in total, not per seed. Combined with the 4 seeds, this yields 29 novel peptides (4 + 25 = 29). The prototypes shown in each panel are included for reference to illustrate the discovery trajectory (Prototype → Seed → Analogs) but are established peptides, not novel discoveries from this work.

---

> ### Author Response · Authors · 2025-11-22
> **Answer to a review (Part 2/2)**
>
> ## Q3: The optimization targeted E. coli, yet validation included Gram-positive strains.
>
> * For PepCompass we deliberately defined a scalar optimization objective as the average log₂ MIC predicted by APEX for three E. coli strains (ATCC 11775, AIC221, AIC222), as stated in Sec. 3.2. This choice was made for two reasons: (i) E.coli is the best-characterized pathogen in literature with the largest number of experimentally validated AMPs; and (ii) it allows to compare PepCompass with prior AMP discovery pipelines that also use E. coli MIC as the main endpoint (see references [4, 20] in manuscript).
> * The wet-lab panel was intentionally broader than the optimization target. We used 14 Gram-negative and 5 Gram-positive strains (Appendix J) to probe two aspects of generalization: (a) whether sequences optimized for E. coli activity retain or improve activity across other Gram-negative pathogens, including MDR isolates; and (b) whether such optimization results in a specialization effect, yielding sequences inactive to Gram-positive bacteria.
>
> As shown in Fig. 9, activity improves consistently from prototypes to seeds to analogs across Gram-negative species, including non-E. coli strains, which supports the view that the E. coli-based objective is a good surrogate for Gram-negatives more broadly.
>
> Because the physicochemical requirements for efficiently targeting Gram-negative outer membranes (LPS-rich, double-membrane architecture) and Gram-positive cell walls (thick peptidoglycan, different surface chemistry) are not identical, it is well-documented that AMPs often exhibit class-specific spectra and that enhancing Gram-negative potency does not automatically translate into enhanced Gram-positive activity. Therefore, the behavior in Fig. 8 for Gram-positive strains is, in our view, the expected specialization effect.
>
> ## Q4: Correlation between the predicted potential and the final wet-lab MIC
>
> To assess how sensitive our discovered seeds are to the accuracy of the surrogate PoGS potential model, we analyzed the relationship between the predicted Log₂ MIC potential and the experimentally validated Log₂ MIC values for all 29 peptides synthesized and tested. As shown in Figure 13, there is a meaningful positive association between the two quantities (Spearman r = 0.31).
>
> Thank you for your constructive and meaningful feedback, which has helped strengthen the manuscript. We hope our responses have addressed your concerns, and would greatly appreciate it if you would consider raising your rating of our submission accordingly.

---

> > ### Comment · Reviewer_i34e · 2025-11-24
> >
> > I appreciate the effort to address all my concerns, particularly those related to the reproducibility and presentation of the wet-lab validation. The paper is significantly strengthened by the rebuttal, so I maintain my positive evaluation.

---

### Official Review · Reviewer_szgr · 2025-10-31

**Soundness:** 2
**Presentation:** 3
**Contribution:** 2
**Rating:** 6
**Confidence:** 3

**Summary:**

This paper introduces PepCompass, a geometry-aware framework for peptide design that leverages Riemannian manifold learning to navigate and optimize peptide embedding spaces. By modeling latent peptide representations as a Union of κ-Stable Riemannian Manifolds, the authors develop three key components: SORBES (a second-order Riemannian Brownian sampler), MUTANG (mutation enumeration in tangent space), and LE-BO/PoGS (local–global Bayesian optimization via potential-minimizing geodesics). The approach enables interpretable and geometry-consistent exploration of peptide space, achieving superior computational and 100% experimental success in antimicrobial peptide discovery. Overall, the work bridges geometric machine learning and biological sequence optimization, offering a mathematically grounded and experimentally validated paradigm for controllable peptide generation.

**Strengths:**

The paper presents a novel and well-structured framework that integrates Riemannian geometry with peptide generative modeling. Its algorithms—SORBES, MUTANG, and LE-BO/PoGS—enable interpretable and geometry-consistent peptide optimization, validated both computationally and experimentally with exceptional success rates. The combination of solid mathematical formulation and real-world biological validation makes the contribution both original and impactful.

**Weaknesses:**

While mathematically elegant, the work’s biological justification for adopting Riemannian geometry remains limited, and some algorithmic components (e.g., κ-stability, second-order correction) lack clear ablation evidence. The framework’s complexity and computational overhead may hinder broader applicability, and its generalization beyond antimicrobial peptides is not yet demonstrated.

**Questions:**

1. How does the proposed Riemannian structure correlate with biologically meaningful properties—e.g., do high-curvature regions correspond to specific physicochemical or functional transitions in peptides?

2. What is the computational scalability of SORBES and LE-BO when applied to longer peptide sequences or larger embedding dimensions?

3. How sensitive is PoGS to the choice of property potential (Φ)? Could it incorporate multiple objectives, such as stability or toxicity, simultaneously?

4. Has PepCompass been tested beyond antimicrobial peptide design (e.g., anticancer or cell-penetrating peptides) to assess its generalizability?

5. How does the Bayesian component in LE-BO handle uncertainty from noisy or biased property predictors, and does it improve robustness in real-world peptide screening?

---

> ### Author Response · Authors · 2025-11-22
> **Answer to a review (Part 1/3)**
>
> We appreciate the time and care you put into your review. We respond to each weakness (W) and question (Q) below, and refer to the updated figures in the resubmitted PDF, Appendix K.
>
> ## W1: Justification for Riemannian geometry, ablations
>
> Our use of Riemannian geometry is supported by ablations comparing against the standard Euclidean alternative. As shown in Table 1 (for PoGS) and Table 2 (for LE-BO), replacing geometric components with Euclidean search consistently degrades performance.
> Additional experiments included in this rebuttal further confirm that these gains are robust across:
>
> * **oracle predictors for LE-BO**:
> To evaluate robustness across different oracles and to verify that additional peptide properties benefit from geometry-aware exploration, we successfully applied LE-BO in three tasks:
>
> #### 1. minimize MIC using DEEP-AMP [1] regressor other than APEX as an oracle
>
> |                      | KY14        | KF16        | KK16        | FL14        | mammuthusin-3 | hydrodamin-2 |
> |----------------------|-------------|-------------|-------------|-------------|----------------|---------------|
> | log₂(MIC) (seed) | 2.00        | 5.06        | 1.85        | 1.82        | 4.02           | 5.09          |
> | Euclidean LE-BO | -0.66 ± 0.50 | 0.04 ± 0.19 | -0.56 ± 0.65 | -0.55 ± 0.46 | -0.48 ± 0.72 | -0.15 ± 0.56 |
> | LE-BO | **-0.80 ± 1.06** | **-0.47 ± 0.38** | **-1.20 ± 1.23** | **-2.35 ± 0.41** | **-0.77 ± 0.75**   | **-0.93 ± 0.89**  |
>
>
> #### 2. minimize toxicity using ToxiPep [2] classification probabilities as an oracle
>
> |                          | KY14        | KF16        | KK16        | FL14        | mammuthusin-3 | hydrodamin-2 |
> |--------------------------|-------------|-------------|-------------|-------------|----------------|---------------|
> | Toxicity (seed)      | 0.8789      | 0.9283      | 0.9044      | 0.9932      | 0.5193         | 0.0522        |
> | Euclidean LE-BO | 0.014 ± 0.002	| 0.016 ± 0.003 | 0.016 ± 0.005 | 0.015 ± 0.001 | 0.014 ± 0.002 | 0.016 ± 0.002 |
> | LE-BO | **0.0115 ± 0.0007** | **0.0117 ± 0.0012** | **0.0120 ± 0.0022** | **0.0122 ± 0.0015** | **0.0126 ± 0.0013** | **0.0135 ± 0.0012** |
>
>
> #### 3. maximize hydrophobicity computed in the Eisenberg scale [3] as an oracle.
>
> |                           | KY14          | KF16          | KK16          | FL14          | mammuthusin-3 | hydrodamin-2 |
> |---------------------------|---------------|---------------|---------------|---------------|----------------|---------------|
> | Hydrophobicity (seed) | -0.34         | -0.60         | 0.03          | 0.12          | 0.17           | -0.33         |
> | Euclidean LE-BO | 1.114 ± 0.091 | 1.208 ± 0.048 | 1.162 ± 0.112 | 1.193 ± 0.029 | 1.323 ± 0.031 | **0.963 ± 0.084** |
> | LE-BO | **1.255 ± 0.013** | **1.270 ± 0.048** | **1.242 ± 0.067** | **1.265 ± 0.026** | **1.325 ± 0.022**  | 0.953 ± 0.095 |
>
>
>
> * **seeds for LE-BO**:
> To show consistent gains compared to ablations across different seeds, we additionally performed an LE-BO ablation study on three further seeds (LL13, RC16, and KI21) from the PoGS optimization. The results confirm that the clear advantage of enabling mutations and of using SORBES-SE as the random walk method instead of Euclidean search can be observed regardless of the chosen seeds.
>
> | Walk        | Mutation | LL13           | RC16           | KI21           | Difference from LE-BO |
> |-------------|----------|----------------|----------------|----------------|------------------------|
> | Euclidean   | x        | 0.787 ± 0.408  | 1.141 ± 0.298  | 1.170 ± 0.283  | 1.033 ± 0.174          |
> | SORBES-SE   | x        | 1.009 ± 0.408  | 1.128 ± 0.298  | 1.120 ± 0.283  | 0.709 ± 0.204          |
> | –           | ✓        | 1.343 ± 0.356  | 1.225 ± 0.275  | 0.803 ± 0.238  | 0.747 ± 0.125          |
> | Euclidean   | ✓        | 0.630 ± 0.190  | 0.751 ± 0.291  | 0.505 ± 0.199  | 0.253 ± 0.091          |
> | SORBES-SE (LE-BO) | ✓  | **0.482 ± 0.226**  | **0.458 ± 0.243**  | **0.190 ± 0.191**  | **0 ± 0**                  |

---

> ### Author Response · Authors · 2025-11-22
> **Answer to a review (Part 2/3)**
>
> * **metrics and prototype sets for PoGS**
> To assess sensitivity of PoGS to the choice of the metric as well as to the choice of prototypes, we compared our implementation with two alternatives: using amino-acid probabilities instead of logits, and straight (Euclidean) interpolation, on a new set of 60 prototypes. Logits yielded better performance in terms of potential, counts of seeds and wells, supporting our choice. Importantly, both metrics outperformed straight interpolation, and PoGS achieved even better results on these new prototypes than those reported in Table 1. These results show that PoGS improves results regardless of the metric used or the prototype set.
>
> | Method                 | Latent Length   | Ambient Length (orig.) | Ambient Length (probs) | Peptide Path Length | Potential          | Seeds      | Wells     |
> |------------------------|------------------|--------------------------|--------------------------|----------------------|---------------------|------------|-----------|
> | Straight interpolation | **6.50 ± 1.34**       | **3448.7 ± 403.8**           | 4.76 ± 4.94              | 47.0 ± 20.9          | -1732.8 ± 272.7     | 117 ± 42   | 41 ± 14   |
> | PoGS (with probs metric) | 11.48 ± 3.36    | 17227.93 ± 147.8         | **1.67 ± 1.16**              | 42.7 ± 18.1          | -1729.9 ± 271.4     | 176 ± 43   | 67 ± 14   |
> | PoGS (original)        | 14.12 ± 3.07      | 5001.4 ± 236.7           | 9.97 ± 4.96              | 63.3 ± 29.3          | **-2329.0 ± 420.0**     | **188 ± 29**   | **80 ± 20**   |
>
> * **κ hyperparameter settings for LE-BO** :
> Our new experiments show that alternative κ values can even outperform those originally selected, demonstrating that LE-BO’s performance can be further enhanced through targeted hyperparameter tuning.
>
> | $\kappa_{mutang}$ | $\kappa_{sorbes}$ | FL14          | KY14          | KF16          | KK16          | mammuthusin-3  | hydrodamin-2  | Avg. Diff. from LE-BO |
> |----------|-----------|----------------|----------------|----------------|----------------|------------------|----------------|-------------------------|
> | 1e-8     | 1e-3      | 0.74 ± 0.28    | 0.83 ± 0.44    | 0.56 ± 0.29    | 0.58 ± 0.30    | **0.44 ± 0.29**      | **0.38 ± 0.13**    | 0.04 ± 0.18             |
> | 1e-4     | 1e-3      | 0.53 ± 0.26    | 0.65 ± 0.39    | 0.66 ± 0.11    | **0.36 ± 0.09**    | 0.60 ± 0.69      | 0.45 ± 0.14    | -0.01 ± 0.12            |
> | 1e-8     | 1e-1      | **0.41 ± 0.11**    | 0.65 ± 0.44    | **0.38 ± 0.14**    | 0.73 ± 0.34    | 0.58 ± 0.21      | 0.45 ± 0.11    | **-0.01 ± 0.19**            |
> | 1e-4     | 1e-1      | 0.83 ± 0.20    | 0.54 ± 0.31    | 0.72 ± 0.29    | 0.66 ± 0.31    | 0.60 ± 0.45      | 0.63 ± 0.40    | 0.12 ± 0.07             |
> | 1e-6 (orig.) | 1e-2 (orig.) | 0.604 ± 0.22 | **0.502 ± 0.24** | 0.600 ± 0.29 | 0.498 ± 0.14 | 0.498 ± 0.38 | 0.581 ± 0.34 | 0.0 ± 0.0 |
>
>
> Second-order corrections are not merely heuristics: Theorem 1 shows they are probably required for convergence to Riemannian Brownian motion. Overall, both theory and experiments (previous and new ablations, superior optimization and 100% wet-lab success) support the biological and algorithmic advantages of our geometry-aware design.
>
> ## W2: Complexity and computational overhead, generalization beyond antimicrobial peptides
> Please refer to answers to Q2, Q3, and Q4 below.
>
> ## Q1: Relation of the Riemannian structure to properties
>
> Our analyses revealed a strong positive correlation between the $\kappa$-stable dimension and peptide length (see Appendix A, Figures 4C,D). This relationship is biologically intuitive: if we interpret the dimension as the number of independent directions in which an object can be modified, then longer peptide sequences naturally offer more positions at which mutations can occur. Our new correlation analyses with physicochemical properties further indicate that local dimensionality is negatively associated with aromaticity and charge (see Appendix K, Figure 11).
> Because computing curvature requires evaluating third-order derivatives of the embedding function - making it computationally prohibitive - we plan to investigate curvature in future work.

---

> ### Author Response · Authors · 2025-11-22
> **Answer to a review (Part 3/3)**
>
> ## Q2: Computational scalability with length and embeddings
>
> Because we use a finite-difference approximation, we can approximate the inference-time cost of Local Enumeration as a function of peptide length or latent dimension. Let the decoder forward-pass time be $t_{decoder}$, the latent dimension k, the peptide length n, and the alphabet size a.
> Then the pessimistic computational cost of the Local Enumeration (LE) steps can be approximated as follows:
> * Computing finite differences for the Jacobian (k evaluations) and the Christoffel symbols (2 evaluations):  (k + 2) * $t_{decoder}$.
>  * SVD computation: O(k * n * a).
>  * Projection of the extrinsic acceleration (second-order correction): O(k * n * a).
>  * MUTANG computation is also O(k * n).
>
> The computational cost of Bayesian Optimization (BO) step in LE-BO is harder to estimate as it depends on the number of LocalEnumeration candidates (affected by $\kappa$-stable dimension of local neighborhoods). However, our empirical runtime evaluation shows that the complete Local Enumeration procedure on the HydrAMP model (k = 64, n = 25, a = 21) takes approximately 1 second per Local Enumeration, and that it is 8x faster than for an alternative Bayesian Optimization-based approach, SAASBO. All computations were performed for the HydrAMP model with dimension 64 and measured on a Mac Mini M4Pro machine with 24GB of RAM.
>
> | Method | Local Enumeration Time / Iteration | Total Iteration Time (s) | Memory (MB) |
> |--------|------------------------------------|----------------------------|-------------|
> | LE-BO  | 1.03 ± 0.31                         | **2.89 ± 2.21**               | **1490 ± 210**  |
> | SAASBO | N/A                                 | 23.31 ± 9.94              | 2248 ± 12   |
>
> ## Q3:Sensitivity of PoGS to property potential, applicability to multiple objectives
> To evaluate PoGS’s sensitivity to the choice of potential and its applicability to multi-objective settings, we applied it to jointly maximize two physicochemical properties by assigning weight α to the hydrophobicity and 1 – α to charge. Across all values of α, PoGS outperformed straight-line (Euclidean) interpolation.For this experiment, we selected a new set of 60 prototypes from the GRAMPA and DRAMP datasets, demonstrating that PoGS achieves superior performance regardless of both the potential used and the prototype set.
>
> | Method                  | Latent Length   | Ambient Length | Peptide Path Length | Max Hydrophobicity | Max Charge     | Maximized Multiobjective Potential |
> |-------------------------|------------------|-----------------|----------------------|----------------------|-----------------|-------------------------------------|
> | PoGS w. α = 0.9     | 12.07 ± 2.57     | 4573.57 ± 19     | 66.86 ± 24.14        | **8.40 ± 0.72**          | **15.81 ± 2.31**    | **24.21 ± 2.42**                        |
> | PoGS w. α = 0.5     | 10.04 ± 2.05     | 3624.69 ± 185.09 | 57.66 ± 27.08        | 7.64 ± 0.63          | 15.75 ± 2.15    | 23.39 ± 2.24                        |
> | PoGS w. α = 0.1     | 10.61 ± 2.13     | 3651.19 ± 190.39 | 55.93 ± 25.80        | 7.91 ± 0.64          | 14.76 ± 2.26    | 22.67 ± 2.35                        |
> | Straight line (Euclidean)| **6.5058 ± 1.34** | **3448.7 ± 403.8** | **47.0 ± 20.9**          | 7.90 ± 0.64          | 13.76 ± 2.22    | 21.66 ± 2.35                        |
>
> ## Q4:Test beyond antimicrobial peptide design
>
> Thanks to your and other Reviewers suggestions, in this rebuttal, we successfully applied PepCompass to alternative properties beyond antimicrobial design, including toxicity (see answer to W1), as well as hydrophobicity and charge (see answer to W1 and Q3).
>
> ## Q5: Bayesian component in LE-BO
> Indeed, we think the Bayesian component helps, as:
> * LE-BO handles noisy or biased predictors via the GP surrogate, which explicitly models predictive uncertainty and steers acquisition toward candidates that remain promising under that uncertainty.
> * Local enumeration restricts search to decoder-consistent neighborhoods, preventing over-optimization on noisy predictor artifacts.
> This improves real-world robustness, as demonstrated by 25/25 LE-BO–designed peptides showing strong in-vitro activity.
>
> Thank you for the thoughtful and constructive feedback, which has significantly improved our manuscript. We believe we have addressed your concerns, and would appreciate it if you considered updating your evaluation accordingly.

---

### Official Review · Reviewer_Dec6 · 2025-11-01

**Soundness:** 3
**Presentation:** 2
**Contribution:** 3
**Rating:** 4
**Confidence:** 5

**Summary:**

The authors proposed PepCompass, a geometry-aware framework for exploring and optimizing antimicrobial peptide (AMP) space. The authors argue that flat Euclidean latent spaces distort distances because they ignore decoder-induced geometry. They formalize a Union of κ-Stable Riemannian Manifolds to capture locally valid geometry and numerical stability, then design two local exploration tools and integrate them into LE-BO, a local-enumeration Bayesian optimization loop. For global search, they introduce PoGS, a potential-minimizing geodesic interpolation biased by predicted antimicrobial activity. The approach reports strong in silico results against diverse baselines and in vitro validation: four PoGS seeds showed activity and LE-BO produced 25/25 highly active peptides across multiple strains, including MDR isolates.

**Strengths:**

The paper motivates why decoder-induced geometry matters and why a single manifold is insufficient, introducing a principled union of κ-stable manifolds with an SVD-based construction to ensure full-rank pullback metrics locally. This directly addresses rank-deficiency in realistic peptide decoders. SORBES includes a second-order correction (Christoffel term) and a convergence guarantee to stopped Riemannian Brownian motion on the κ-stable chart (Theorem 1). A practical finite-difference variant (SORBES-SE) with adaptive steps is also provided. MUTANG turns tangent directions into concrete amino-acid substitutions and enumerates local neighborhoods in sequence space—bridging continuous geometry and discrete edits. LE-BO leverages local enumeration rather than latent Euclidean kernels, with ROBOT-style diversity for exploration. PoGS adds a property-aware geodesic objective that yields more seeds/wells than straight interpolation.

**Weaknesses:**

The paper itself notes a limitation—using Euclidean distance on decoder logits as the ambient metric for PoGS; while the authors argue it’s a stable first-order surrogate, alternative metrics might better match biochemical structure or synthesis constraints. A sensitivity analysis to this metric would strengthen claims.  While SORBES-SE mitigates cost, repeatedly estimating κ-stable charts, Christoffel terms (even indirectly), and re-encoding during LOCALENUMERATION could be heavy for large-scale campaigns; clearer runtime/memory profiling beyond the remark that other BO baselines were too slow would help assess deployment trade-offs.Many gains depend on the chosen property predictor (e.g., APEX MIC regressor) and seed selection from PoGS. More detail on robustness to different oracles, calibration under domain shift, and ablations on κ/α thresholds would help generalize the conclusions.

**Questions:**

1. You motivate unions of manifolds by decoder rank deficiency. Could you quantify how often and how severely the rank drops across the latent space of your trained peptide models, and how this correlates with optimization failure modes under Euclidean search?
2. Beyond antimicrobial activity, what other peptide properties (e.g., hemolysis, proteolytic stability, solubility) most benefit from geometry-aware exploration, and why would Euclidean search be particularly misleading for them?
3. In Table 2, LE-BO dominates most baselines. Which single design decision (SORBES vs. MUTANG vs. ROBOT diversity) contributed most to the gains, according to your ablations, and is that consistent across different seeds?

---

> ### Author Response · Authors · 2025-11-22
> **Answer to a review (Part 1/3)**
>
> Thank you for taking the time to provide such an insightful review. We address each weakness (W) and question (Q) below. For figures, please refer to resubmitted pdf, Appendix K.
>
> ## W1: Sensitivity analysis of PoGS to metrics
>
> To assess sensitivity of PoGS to the choice of the metric as well as to the choice of prototypes, we compared our implementation with two alternatives: using amino-acid probabilities instead of logits, and straight (Euclidean) interpolation, on a new set of 60 prototypes. Logits yielded better performance in terms of potential, counts of seeds and wells, supporting our choice. Importantly, both metrics outperformed straight interpolation, and PoGS achieved even better results on these new prototypes than those reported in Table 1. These results show that PoGS improves results regardless of the metric used or the prototype set.
>
> | Method                 | Latent Length   | Ambient Length (orig.) | Ambient Length (probs) | Peptide Path Length | Potential          | Seeds      | Wells     |
> |------------------------|------------------|--------------------------|--------------------------|----------------------|---------------------|------------|-----------|
> | Straight interpolation | **6.50 ± 1.34**       | **3448.7 ± 403.8**           | 4.76 ± 4.94              | 47.0 ± 20.9          | -1732.8 ± 272.7     | 117 ± 42   | 41 ± 14   |
> | PoGS (with probs metric) | 11.48 ± 3.36    | 17227.93 ± 147.8         | **1.67 ± 1.16**              | 42.7 ± 18.1          | -1729.9 ± 271.4     | 176 ± 43   | 67 ± 14   |
> | PoGS (original)        | 14.12 ± 3.07      | 5001.4 ± 236.7           | 9.97 ± 4.96              | 63.3 ± 29.3          | **-2329.0 ± 420.0**     | **188 ± 29**   | **80 ± 20**   |
>
> To be integrated into our framework, a metric must operate on the continuous decoder-output space and be differentiable. This excludes existing structure- or physicochemical–based metrics, which do not meet these criteria. Developing differentiable, biochemically informed metrics is an important direction for future work.
>
> ## W2: Clearer runtime/memory profiling
>
> To quantify the computational gains of our method, we measured the average runtime of LE-BO over 10 iterations and 6 seeds, and compared it to SAASBO under the same conditions. LE-BO required 8× less time per iteration and used 1.5× less memory. Moreover, Local Enumeration accounted for only 35% of the total runtime, underscoring its efficiency. Average execution time of a single optimization run of LE-Bo with 1400 iterations was 1h20m (±30m). All computations were performed for the HydrAMP model with dimension 64 and measured on a Mac Mini M4Pro machine with 24GB of RAM.
>
> | Method | Local Enumeration Time / Iteration | Total Iteration Time (s) | Memory (MB) |
> |--------|------------------------------------|----------------------------|-------------|
> | LE-BO  | 1.03 ± 0.31                         | **2.89 ± 2.21**               | **1490 ± 210**  |
> | SAASBO | N/A                                 | 23.31 ± 9.94              | 2248 ± 12   |
>
> ## W3: More detail on robustness to hyperparameters and oracles
> We are grateful to the Reviewer for the suggestion of using different hyperparameters! Our new experiments show that alternative $κ$ values can even outperform those originally selected, demonstrating that LE-BO’s performance can be further enhanced through targeted hyperparameter tuning. An ablation with respect to α is unnecessary because, with only a single SORBES-SE step, α has no effect on the search process.
> For robustness to different oracles, please refer to Q2.
>
> | $κ_{mutang}$ | $κ_{sorbes}$ | FL14          | KY14          | KF16          | KK16          | mammuthusin-3  | hydrodamin-2  | Avg. Diff. from LE-BO |
> |----------|-----------|----------------|----------------|----------------|----------------|------------------|----------------|-------------------------|
> | 1e-8     | 1e-3      | 0.74 ± 0.28    | 0.83 ± 0.44    | 0.56 ± 0.29    | 0.58 ± 0.30    | **0.44 ± 0.29**      | **0.38 ± 0.13**    | 0.04 ± 0.18             |
> | 1e-4     | 1e-3      | 0.53 ± 0.26    | 0.65 ± 0.39    | 0.66 ± 0.11    | **0.36 ± 0.09**    | 0.60 ± 0.69      | 0.45 ± 0.14    | -0.01 ± 0.12            |
> | 1e-8     | 1e-1      | **0.41 ± 0.11**    | 0.65 ± 0.44    | **0.38 ± 0.14**    | 0.73 ± 0.34    | 0.58 ± 0.21      | 0.45 ± 0.11    | **-0.01 ± 0.19**            |
> | 1e-4     | 1e-1      | 0.83 ± 0.20    | 0.54 ± 0.31    | 0.72 ± 0.29    | 0.66 ± 0.31    | 0.60 ± 0.45      | 0.63 ± 0.40    | 0.12 ± 0.07             |
> | 1e-6 (orig.) | 1e-2 (orig.) | 0.604 ± 0.22 | **0.502 ± 0.24** | 0.600 ± 0.29 | 0.498 ± 0.14 | 0.498 ± 0.38 | 0.581 ± 0.34 | 0.0 ± 0.0 |
>
> (to be continued)

---

> ### Author Response · Authors · 2025-11-22
> **Answer to a review (Part 2/3)**
>
> To evaluate robustness across different oracles and to verify that additional peptide properties benefit from geometry-aware exploration, we successfully applied LE-BO in three tasks:
>
> #### 1. minimize MIC using DEEP-AMP [1] regressor other than APEX as an oracle
>
> |                      | KY14        | KF16        | KK16        | FL14        | mammuthusin-3 | hydrodamin-2 |
> |----------------------|-------------|-------------|-------------|-------------|----------------|---------------|
> | log₂(MIC) (seed) | 2.00        | 5.06        | 1.85        | 1.82        | 4.02           | 5.09          |
> | Euclidean LE-BO | -0.66 ± 0.50 | 0.04 ± 0.19 | -0.56 ± 0.65 | -0.55 ± 0.46 | -0.48 ± 0.72 | -0.15 ± 0.56 |
> | LE-BO | **-0.80 ± 1.06** | **-0.47 ± 0.38** | **-1.20 ± 1.23** | **-2.35 ± 0.41** | **-0.77 ± 0.75**   | **-0.93 ± 0.89**  |
>
> #### 2. minimize toxicity using ToxiPep [2] classification probabilities as an oracle
>
> |                          | KY14        | KF16        | KK16        | FL14        | mammuthusin-3 | hydrodamin-2 |
> |--------------------------|-------------|-------------|-------------|-------------|----------------|---------------|
> | Toxicity (seed)      | 0.8789      | 0.9283      | 0.9044      | 0.9932      | 0.5193         | 0.0522        |
> | Euclidean LE-BO | 0.014 ± 0.002	| 0.016 ± 0.003 | 0.016 ± 0.005 | 0.015 ± 0.001 | 0.014 ± 0.002 | 0.016 ± 0.002 |
> | LE-BO | **0.0115 ± 0.0007** | **0.0117 ± 0.0012** | **0.0120 ± 0.0022** | **0.0122 ± 0.0015** | **0.0126 ± 0.0013** | **0.0135 ± 0.0012** |
>
> #### 3. maximize hydrophobicity computed in the Eisenberg scale [3] as an oracle.
>
> |                           | KY14          | KF16          | KK16          | FL14          | mammuthusin-3 | hydrodamin-2 |
> |---------------------------|---------------|---------------|---------------|---------------|----------------|---------------|
> | Hydrophobicity (seed) | -0.34         | -0.60         | 0.03          | 0.12          | 0.17           | -0.33         |
> | Euclidean LE-BO | 1.114 ± 0.091 | 1.208 ± 0.048 | 1.162 ± 0.112 | 1.193 ± 0.029 | 1.323 ± 0.031 | **0.963 ± 0.084** |
> | LE-BO | **1.255 ± 0.013** | **1.270 ± 0.048** | **1.242 ± 0.067** | **1.265 ± 0.026** | **1.325 ± 0.022**  | 0.953 ± 0.095 |
>
> ## Q1: Rank deficiency quantification, inefficiency of Euclidean search
>
> As evaluated for two models, HydrAMP and PepCVAE, the effective rank is almost always much lower than the latent dimension (see distribution of the effective rank quantified in Figures 4A, B). Moreover, it strongly positively correlates with peptide length (see Figures 4 C,D in Appendix A).
>
> Our new analysis in Figure 10 in Appendix K shows that LE-BO with SORBES-SE and mutation enabled (Fig 10 A, E) identifies more candidates in Local Enumeration for higher $\kappa$-stable dimensions / longer peptides than for lower dimensions / shorter peptides. Euclidean-based search exhibits the opposite trend (Fig. 10, B,C, F,G). As our optimization method relies on dense populations of peptide neighborhoods, this explains weaker performance of Euclidean approaches.
> Moreover, LE-BO with SORBES-SE and mutation produces substantially more candidates on average (16 730 ± 14 786; Fig. 10 A, E) than the Euclidean search with mutation (442 ± 151; Fig. 10 B, F). Even without mutation, SORBES-SE yields higher candidate counts (17 ± 5; Fig. 10, Panels D, H) than its Euclidean counterpart (13 ± 5; Fig. 10, Panels C, G).
>
> ## Q2: Other peptide properties benefitting from geometry-aware exploration compared to Euclidean
>
> We appreciate the insights gained thanks to your question! Our new analysis (Figure 11 in Appendix K) revealed that the κ-stable dimension of the latent space is strongly negatively correlated with charge (Spearman r = -0.55) and aromaticity (r = -0.57). This indicates that peptides with high latent dimension tend to have higher solubility, but lower membrane affinity and antimicrobial activity. As discussed in response to Q2, LE-BO identifies substantially more candidates than Euclidean methods in these high-dimensional regions, increasing the likelihood that the unfavorable antimicrobial properties can be corrected.
>
> ## Q3. Ablations without ROBOT & confirmation across seeds
>
> Thank you for your suggestion! Our new ablation analysis confirms that ROBOT improves LE-BO for 4 out of 6 seeds (with average difference in log MIC of -0.03).
>
> | Method            | FL14          | KY14          | KF16          | KK16          | mammuthusin-3 | hydrodamin-2 | Avg. diff. from LE-BO |
> |-------------------|---------------|---------------|---------------|---------------|----------------|---------------|-------------------------|
> |*LE-BO w/o ROBOT | 0.75 ± 0.41   | 0.52 ± 0.17   | **0.56 ± 0.11**   | 0.63 ± 0.28   | 0.57 ± 0.61     | **0.42 ± 0.20**   | 0.03 ± 0.12            |
> | LE-BO          | **0.604 ± 0.22**  | **0.502 ± 0.24**  | 0.600 ± 0.29  | **0.498 ± 0.14**  | **0.498 ± 0.38**    | 0.581 ± 0.34  | **0 ± 0**                  |
>
> (to be continued)

---

> ### Author Response · Authors · 2025-11-22
> **Answer to a review (Part 3/3)**
>
> To show consistent gains compared to ablations across different seeds, we additionally performed an LE-BO ablation study on three further seeds (LL13, RC16, and KI21) from the PoGS optimization. The results confirm that clear advantage of enabling mutations and of using SORBES-SE as the random walk method can be observed regardless of the chosen seeds. Taken together, all three LE-BO components - SORBES-SE, enabling mutation (MUTANG), and ROBOT - contribute meaningful improvements. However, enabling mutation with MUTANG provides the most substantial benefit: across all ablations, the LE-BO variants incorporating MUTANG consistently outperform their counterparts without it.
>
> | Walk        | Mutation | LL13           | RC16           | KI21           | Difference from LE-BO |
> |-------------|----------|----------------|----------------|----------------|------------------------|
> | Euclidean   | x        | 0.787 ± 0.408  | 1.141 ± 0.298  | 1.170 ± 0.283  | 1.033 ± 0.174          |
> | SORBES-SE   | x        | 1.009 ± 0.408  | 1.128 ± 0.298  | 1.120 ± 0.283  | 0.709 ± 0.204          |
> | –           | ✓        | 1.343 ± 0.356  | 1.225 ± 0.275  | 0.803 ± 0.238  | 0.747 ± 0.125          |
> | Euclidean   | ✓        | 0.630 ± 0.190  | 0.751 ± 0.291  | 0.505 ± 0.199  | 0.253 ± 0.091          |
> | SORBES-SE (LE-BO) | ✓  | **0.482 ± 0.226**  | **0.458 ± 0.243**  | **0.190 ± 0.191**  | **0 ± 0**                  |
>
> Thank you for the thoughtful and constructive comments, which greatly helped us improve the manuscript. We hope we have addressed your concerns, and if so, we would be grateful if you would consider raising your rating of our submission.
>
> References
>
> [1] Pandi et al. Nat Commun. 2023, 10.1038/s41467-023-42434-9
>
> [2] Guan et al. Comput. Struct. Biotechnol. J., 2025.10.1016/j.csbj.2025.05.039
>
> [3] Eisenberg et al.,J Mol Biol, 1984

---

### Author Response · Authors · 2025-11-29
**Author final remarks / rebuttal summary**

We are grateful to the reviewers for their thoughtful and insightful comments, and to the previous and the new Area Chair for orchestrating the review process. Given the unusual rebuttal constraints posed by the ICLR 2026 data leak, we provide this concise summary to help facilitate the final evaluation and ensure that the core clarifications and new results are easy to review.

---

# Summary of Reviewer Comments and Our Responses

## 1. Sensitivity Analyses to Metrics and Hyperparameters

Reviewers asked whether performance depends heavily on design choices such as metrics and hyperparameters.

**Key findings:**
* **PoGS is robust** across metrics (logits, AA-probabilities, Euclidean) and across 60 new prototypes.
* PoGS with differentiable metrics consistently **outperform Euclidean interpolation**, with the chosen metric being strongest.
* While LE-BO already delivers best-in-class performance across κ values, tuning κ can yield additional gains.
* PoGS remains robust under various **single and multi-objective potentials**, always surpassing Euclidean interpolation.

---

## 2. Justification for Geometry-Aware Methods (PoGS & LE-BO)

**Key findings:**
* Replacing any geometric component with a Euclidean one causes **consistent performance degradation** for PoGS and LE-BO.
* Experiments across seeds, metrics, potentials, and tasks confirm that **Riemannian search identifies more candidates**, especially in high-dimensional latent regions.
---

## 3. Runtime, Memory, and Scalability

**Key findings:**
- **LE-BO is ~8× faster** per iteration than SAASBO and uses **~1.5× less memory**.
- Local Enumeration accounts for only ~35% of total runtime.
- A full **1400-iteration run takes ~1h 20m ± 30m** on a Mac Mini M4 Pro (24GB RAM).
- Runtime scales mainly with decoder calls; Local Enumeration remains **~1s/step** even for k=64, n=25.

---

## 4. Robustness Across Oracles and Tasks

We added three new oracle tasks:
1. **Minimize MIC using DEEP-AMP**
2. **Minimize toxicity using ToxiPep**
3. **Maximize hydrophobicity**

**Key findings:**
- LE-BO makes strong improvements across all tasks.
- **Geometry-aware LE-BO consistently outperforms Euclidean LE-BO** on *every task*.

---

## 5. Biological Interpretation of Latent Geometry

**Key findings:**
- **$\kappa$-k-stable dimension correlates strongly with peptide length**, matching biological intuition.
* Negative correlations with **charge** and **aromaticity** indicate high-dimensional latent regions correspond to peptides with higher solubility and lower membrane affinity.
* Geometry-aware search explores these regions **more effectively than Euclidean methods**, increasing chances of improving anti-microbial properties.

---

## 6. Ablations and Component Contributions (ROBOT, MUTANG, SORBES-SE)

**Key findings:**

* **Mutation with MUTANG gives the largest gains**.
* **SORBES-SE-based methods consistently produce a higher number of candidates than Euclidean counterparts.**
* **ROBOT** provides smaller but consistent gains.
* Across new seeds (LL13, RC16, KI21), **full LE-BO always performs best.**

---

# Overall Conclusions

* **Riemannian geometry is both theoretically justified and empirically superior** to Euclidean alternatives across PoGS and LE-BO.
* **LE-BO is computationally efficient**, with strong runtime and memory advantages.
* **Robust performance generalizes broadly** across metrics, seeds, prototype sets, latent potentials, and oracle predictors.
* **Ablations verify that all components matter**, especially MUTANG and Riemannian corrections.
* **Latent geometry correlates with meaningful biological properties**, supporting interpretability and design rationale.
* **To support transparency and reproducibility, we have publicly released all peptide sequences discovered during our experiments.**
---
Overall, we believe this work will make a meaningful contribution to the ICLR community, particularly in the areas of geometry-aware autoencoders and AI-driven peptide design.

---

### Note · Program_Chairs · 2026-01-17
**Submission Desk Rejected by Program Chairs**

The following references in this submission do not refer to real documents and/or have major errors in bibliographic information:

     Steffen Schneider et al. Internal representations of vision models through the lens of frames on data manifolds. arXiv preprint, 2022. URL https://arxiv.org/abs/2211.10558.
    Seonghyeon Park et al. Low-rank subspaces in gans. In NeurIPS, 2021.